# SDAC: Efficient Safe Reinforcement Learning with Low-Bias Distributional Actor-Critic

## Abstract

To apply reinforcement learning (RL) to real-world applications, agents are required to adhere to the safety guidelines of their respective domains. Safe RL can effectively handle the guidelines by converting them into constraints of the RL problem. In this paper, we develop a safe distributional RL method based on the trust region method, which can satisfy constraints consistently. However, policies may not meet the safety guidelines due to the estimation bias of distributional critics, and importance sampling required for the trust region method can hinder performance due to its significant variance. Hence, we enhance safety performance through the following approaches. First, we train distributional critics to have low estimation biases using proposed target distributions where bias-variance can be traded off. Second, we propose novel surrogates for the trust region method expressed with Q-functions using the reparameterization trick. Additionally, depending on initial policy settings, there can be no policy satisfying constraints within a trust region. To handle this infeasible issue, we propose a gradient integration method which guarantees to find a policy satisfying all constraints from an unsafe initial policy. From extensive experiments, the proposed method with risk-averse constraints shows minimal constraint violations while achieving high returns compared to existing safe RL methods. Furthermore, we demonstrate the benefit of safe RL for problems in which the reward cannot be easily specified.

## 1 Introduction

Deep reinforcement learning (RL) enables reliable control of complex robots (Merel et al., 2020; Peng et al., 2021; Rudin et al., 2022). Miki et al. (2022) have shown that RL can control quadrupedal robots more robustly than existing model-based optimal control methods, and Peng et al. (2022) have performed complex natural motion tasks using physically simulated characters. In order to successfully apply RL to real-world systems, it is essential to design a proper reward function which reflects safety guidelines, such as collision avoidance and limited energy consumption, as well as the goal of the given task. However, finding the reward function that considers all of such factors involves a cumbersome and time-consuming task since RL algorithms must be repeatedly performed to verify the results of the designed reward function. Instead, *safe RL*, which handles safety guidelines as constraints, can be an appropriate solution. A safe RL problem can be formulated using a constrained Markov decision process (Altman, 1999), where not only the reward but also cost functions, which output the safety guideline signals, are defined. By defining constraints using risk measures, such as condtional value at risk (CVaR), of the sum of costs, safe RL aims to maximize returns while satisfying the constraints. Under the safe RL framework, the training process becomes straightforward since there is no need to search for a reward that reflects the safety guidelines.

The most crucial part of safe RL is to satisfy the safety constraints, and it requires two conditions. First, constraints should be estimated with low biases. In general RL, the return is estimated using a function estimator called a critic, and, in safe RL, additional critics are used to estimate the constraint values. In our case, constraints are defined using risk measures, so it is essential to use distributional critics (Dabney et al., 2018b). Then, the critics can be trained using the distributional Bellman update (Bellemare et al., 2017). However, the Bellman update only considers the one-step temporal difference, which can induce a large bias. The estimation bias makes it difficult for critics to judge the policy, which can lead to the policy becoming overly conservative or risky, as shown in Section 5.3. Therefore, there is a need for a method that can train distributional critics with low biases.

Second, a policy update method considering safety constraints, denoted by a *safe policy update rule*, is required not only to maximize the reward sum but also to satisfy the constraints after updating the policy. Existing safe policy update rules can be divided into the trust region-based and Lagrangian methods. The trust region-based method calculates the update direction by approximating the safe RL problem within a trust region and updates the policy through a line search (Yang et al., 2020; Kim & Oh, 2022a). The Lagrangian method converts the safe RL problem into a dual problem and updates the policy and Lagrange multipliers (Yang et al., 2021). However, the Lagrangian method is difficult to guarantee satisfying constraints during training theoretically, and the training process can be unstable due to the multipliers (Stooke et al., 2020). In contrast, trust region-based methods can guarantee to improve returns while satisfying constraints under tabular settings (Achiam et al., 2017). Still, trust region-based methods also have critical issues. There can be an infeasible starting case, meaning that no policy satisfies constraints within the trust region due to initial policy settings. Thus, proper handling of this case is required, but there is a lack of such handling methods when there are multiple constraints. Furthermore, the trust region-based methods are known as not sample-efficient, as observed in several RL benchmarks (Achiam, 2018; Raffin et al., 2021).

In this paper, we propose an efficient trust region-based safe RL algorithm with multiple constraints, called a *safe distributional actor-critic* (SDAC). First, to train critics to estimate constraints with low biases, we propose a *TD($\lambda$) target distribution* combining multiple-step distributions, where bias-variance can be traded off by adjusting the trace-decay $\lambda$. Then, under off-policy settings, we present a memory-efficient method to approximate the TD($\lambda$) target distribution using quantile distributions (Dabney et al., 2018b), which parameterize a distribution as a sum of Dirac functions. Second, to handle the infeasible starting case for multiple constraint settings, we propose a *gradient integration method*, which recovers policies by reflecting all constraints simultaneously. It guarantees to obtain a policy which satisfies the constraints within a finite time under mild technical assumptions. Also, since all constraints are reflected at once, it can restore the policy more stably than existing handling methods Xu et al. (2021), which consider only one constraint at a time. Finally, to improve the efficiency of the trust region method as much as Soft Actor-Critic (SAC) (Haarnoja et al., 2018), we propose novel *SAC-style surrogates*. We show that the surrogates have bounds within a trust region and empirically confirm improved efficiency in Appendix B. In summary, the proposed algorithm trains distributional critics with low biases using the TD($\lambda$) target distributions and updates a policy using safe policy update rules with the SAC-style surrogates. If the policy cannot satisfy constraints within the trust region, the gradient integration method recovers the policy to a feasible policy set.

To evaluate the proposed method, we conduct extensive experiments with four tasks in the Safety Gym environment (Ray et al., 2019) and show that the proposed method with risk-averse constraints achieves high returns with minimal constraint violations during training compared to other safe RL baselines. Also, we experiment with locomotion tasks using robots with different dynamic and kinematic models to demonstrate the advantage of safe RL over traditional RL, such as no reward engineering required. The proposed method has successfully trained locomotion policies with the same straightforward reward and constraints for different robots with different configurations.

## 2 BACKGROUND

**Constrained Markov Decision Processes.** We formulate the safe RL problem using constrained Markov decision processes (CMDPs) (Altman, 1999). A CMDP is defined as $(S, A, P, R, C_{1,..,K}, \rho, \gamma)$, where $S$ is a state space, $A$ is an action space, $P : S \times A \times S \mapsto [0, 1]$ is a transition model, $R : S \times A \times S \mapsto \mathbb{R}$ is a reward function, $C_{k \in \{1,...,K\}} : S \times A \times S \mapsto \mathbb{R}_{\geq 0}$ are cost functions, $\rho : S \mapsto [0, 1]$ is an initial state distribution, and $\gamma \in (0, 1)$ is a discount factor. The state action value, state value, and advantage functions are defined as follows:

$$Q_R^\pi(s, a) := \mathop{\mathbb{E}}_{\pi, P} \left[ \sum_{t=0}^\infty \gamma^t R(s_t, a_t, s_{t+1}) \Big| s_0 = s, a_0 = a \right],$$

$$V_R^\pi(s) := \mathop{\mathbb{E}}_{\pi, P} \left[ \sum_{t=0}^\infty \gamma^t R(s_t, a_t, s_{t+1}) \Big| s_0 = s \right], A_R^\pi(s, a) := Q_R^\pi(s, a) - V_R^\pi(s). \tag{1}$$

By substituting the costs for the reward, the cost value functions $V_{C_k}^\pi(s), Q_{C_k}^\pi(s, a), A_{C_k}^\pi(s, a)$ are defined. In the remainder of the paper, the cost parts will be omitted since they can be retrieved by replacing the reward with the costs. Given a policy $\pi$ from a stochastic policy set $\Pi$, the discounted

state distribution is defined as $d^\pi(s) := (1 - \gamma) \sum_{t=0}^\infty \gamma^t \Pr(s_t = s|\pi)$, and the return is defined as $Z_R^\pi(s, a) := \sum_{t=0}^\infty \gamma^t R(s_t, a_t, s_{t+1})$, where $s_0 = s$, $a_0 = a$, $a_t \sim \pi(\cdot|s_t)$, and $s_{t+1} \sim P(\cdot|s_t, a_t)$. Then, the safe RL problem is defined as follows with a safety measure $F$:

$$\max_\pi \mathbb{E}\left[Z_R^\pi(s, a)|s \sim \rho, a \sim \pi(\cdot|s)\right] \text{ s.t. } F(Z_{C_k}^\pi(s, a)|s \sim \rho, a \sim \pi(\cdot|s)) \leq d_k \; \forall k, \qquad (2)$$

where $d_k$ is a limit value of the $k$th constraint.

**Mean-Std Constraints.** In our safe RL setting, we use mean-std as the safety measure: $F(Z; \alpha) = \mathbb{E}[Z] + (\phi(\Phi^{-1}(\alpha))/\alpha) \cdot \text{Std}[Z]$, where $\alpha \in (0, 1]$ adjusts conservativeness of constraints, $\text{Std}[Z]$ is the standard deviation of $Z$, $\phi$ is the probability density function, and $\Phi$ is the cumulative distribution function (CDF) of the standard normal distribution. The mean-std is identical to the conditional value at risk (CVaR) if $Z$ follows the Gaussian distribution, and the mean-std constraint can effectively reduce the number of constraint violations, as shown by Yang et al. (2021); Kim & Oh (2022b;a). To estimate the mean-std of cost returns, Kim & Oh (2022b) define the square value functions: $S_{C_k}^\pi(s) := \mathop{\mathbb{E}}_{\pi, P}\left[Z_{C_k}^\pi(s, a)^2|a \sim \pi(\cdot|s)\right], S_{C_k}^\pi(s, a) := \mathop{\mathbb{E}}_{\pi, P}\left[Z_{C_k}^\pi(s, a)^2\right]$, and $A_{S_k}^\pi(s, a) := S_{C_k}^\pi(s, a) - S_{C_k}^\pi(s)$. Additionally, $d_2^\pi(s) := (1 - \gamma^2) \sum_{t=0}^\infty \gamma^{2t} \Pr(s_t = s|\pi)$ denotes a doubly discounted state distribution. Then, the $k$th constraint can be written as follows:

$$F_k(\pi; \alpha) = J_{C_k}(\pi) + \frac{\phi(\Phi^{-1}(\alpha))}{\alpha}\sqrt{J_{S_k}(\pi) - J_{C_k}(\pi)^2} \leq d_k, \qquad (3)$$

where $J_{C_k}(\pi) := \mathop{\mathbb{E}}_{s \sim \rho}\left[V_{C_k}^\pi(s)\right]$ and $J_{S_k}(\pi) := \mathop{\mathbb{E}}_{s \sim \rho}\left[S_{C_k}^\pi(s)\right]$.

**Distributional Quantile Critic.** To parameterize the distribution of the returns, Dabney et al. (2018b) have proposed an approximation method to estimate the returns using the following quantile distributions, called a *distributional quantile critic*: $\Pr(Z_{R,\theta}^\pi(s, a) = z) := \sum_{m=1}^M \delta_{\theta_m(s,a)}(z)/M$, where $M$ is the number of atoms, $\theta$ is a parametric model, and $\theta_m(s, a)$ is the $m$th atom. The percentile value of the $m$th atom is denoted by $\tau_m$ ($\tau_0 = 0, \tau_i = i/M$). In distributional RL, the returns are directly estimated to get value functions, and the target distribution can be calculated from the distributional Bellman operator (Bellemare et al., 2017): $\mathcal{T}^\pi Z_R(s, a) \overset{D}{:=} R(s, a, s') + \gamma Z_R(s', a')$, where $s' \sim P(\cdot|s, a)$ and $a' \sim \pi(\cdot|s')$. The above one-step operator can be expanded to the $n$-step one: $\mathcal{T}_n^\pi Z_R(s_0, a_0) \overset{D}{:=} \sum_{t=0}^{n-1} \gamma^t R(s_t, a_t, s_{t+1}) + \gamma^n Z_R(s_n, a_n)$. Then, the critic can be trained to minimize the following quantile regression loss (Dabney et al., 2018b):

$$\mathcal{L}(\theta) = \sum_{m=1}^M \underbrace{\mathbb{E}_{\bar{Z} \sim Z}\left[\rho_{\hat{\tau}_m}(\bar{Z} - \theta_m)\right]}_{=:\mathcal{L}_{\text{QR}}^{\hat{\tau}_m}(\theta_m)}, \text{ where } \rho_\tau(x) = x \cdot (\tau - \mathbf{1}_{\text{x}<0}), \; \hat{\tau}_m := \frac{\tau_{m-1} + \tau_m}{2}, \qquad (4)$$

and $\mathcal{L}_{\text{QR}}^\tau(\theta)$ denotes the quantile regression loss for a single atom. The distributional quantile critic can be plugged into existing actor-critic algorithms because only the critic modeling is changed.

## 3 PROPOSED METHOD

We propose the following three approaches to enhance the safety performance of trust region-based safe RL methods. First, we introduce a TD($\lambda$) target distribution combining $n$-step distributions, which can trade off bias-variance. The target distribution enables training of the distributional critics with low biases. Second, we propose novel surrogate functions for policy updates that empirically improve the performance of the trust region method. Finally, we present a gradient integration method under multiple constraint settings to handle the infeasible starting cases.

### 3.1 TD($\lambda$) TARGET DISTRIBUTION

In this section, we propose a target distribution by capturing that the TD($\lambda$) loss, which is obtained by a weighted sum of several losses, and the quantile regression loss with a single distribution are equal. A recursive method is then introduced so that the target distribution can be obtained practically. First, the $n$-step targets are estimated as follows, after collecting trajectories $(s_t, a_t, s_{t+1}, ...)$ with a behavioral policy $\mu$:

$$\hat{Z}_t^{(n)} \overset{D}{:=} R_t + \gamma R_{t+1} + \gamma^2 R_{t+2} + \cdots + \gamma^{n-1} R_{t+n-1} + \gamma^n Z_{R,\theta}^\pi(s_{t+n}, a'_{t+n}), \qquad (5)$$

Figure 1: Constructing procedure for target distribution. First, multiply the target at $t+1$ step by $\gamma$ and add $R_t$. Next, weight-combine the shifted previous target and one-step target at $t$ step and restore the CDF of the combined target. The CDF can be restored by sorting the positions of the atoms and then accumulating the weights at each atom position. Finally, the projected target can be obtained by finding the positions of the atoms corresponding to $M'$ quantiles in the CDF. Using the projected target, the target at $t-1$ step can be found recursively.

where $R_t = R(s_t, a_t, s_{t+1})$, $a'_{t+n} \sim \pi(\cdot|s_{t+n})$, and $\pi$ is the current policy. Note that the $n$-step target controls the bias-variance tradeoff using $n$. If $n$ is equal to 1, the $n$-step target is equivalent to the temporal difference method that has low variance but high bias. On the contrary, if $n$ goes to infinity, it becomes a Monte-Carlo estimation that has high variance but low bias. However, finding proper $n$ is another cumbersome task. To alleviate this issue, TD($\lambda$) (Sutton, 1988) method considers the discounted sum of all $n$-step targets. Similar to TD($\lambda$), we define the TD($\lambda$) loss for the distributional quantile critic as the discounted sum of all quantile regression losses with $n$-step targets. Then, the TD($\lambda$) loss for a single atom is approximated using importance sampling of the sampled $n$-step targets in (5) as:

$$\mathcal{L}_{\mathrm{QR}}^{\tau}(\theta) = (1-\lambda) \sum_{i=0}^{\infty} \lambda^i \mathbb{E}_{\bar{Z} \sim \mathcal{T}_i^{\pi} Z} \left[ \rho_\tau(\bar{Z}-\theta) \right] \approx \frac{1-\lambda}{M} \sum_{i=0}^{\infty} \lambda^i \prod_{j=1}^{i} \frac{\pi(a_{t+j}|s_{t+j})}{\mu(a_{t+j}|s_{t+j})} \sum_{m=1}^{M} \rho_\tau(\hat{Z}_{t,m}^{(i+1)} - \theta), \quad (6)$$

where $\lambda$ is a trace-decay value, and $\hat{Z}_{t,m}^{(i)}$ is the $m$th atom of $\hat{Z}_t^{(i)}$. Since $\hat{Z}_t^{(i)} \overset{D}{=} R_t + \gamma \hat{Z}_{t+1}^{(i-1)}$ is satisfied, (6) is the same as the quantile regression loss with the following single distribution $\hat{Z}_t^{\mathrm{tot}}$, called a *TD($\lambda$) target distribution*:

$$\Pr(\hat{Z}_t^{\mathrm{tot}} = z) := \frac{1}{\mathcal{N}} \frac{1-\lambda}{M} \sum_{i=0}^{\infty} \lambda^i \prod_{j=1}^{i} \frac{\pi(a_{t+j}|s_{t+j})}{\mu(a_{t+j}|s_{t+j})} \sum_{m=1}^{M} \delta_{\hat{Z}_{t,m}^{(i+1)}}(z)$$

$$= \frac{1}{\mathcal{N}} \left\{ (1-\lambda) \underbrace{\sum_{m=1}^{M} \frac{1}{M} \delta_{\hat{Z}_{t,m}^{(1)}}(z)}_{\text{One step TD target}} + \lambda \frac{\pi(a_{t+1}|s_{t+1})}{\mu(a_{t+1}|s_{t+1})} \underbrace{\Pr(R_t + \gamma \hat{Z}_{t+1}^{\mathrm{tot}} = z)}_{\text{Previous TD($\lambda$) target}} \right\}, \quad (7)$$

where $\mathcal{N}$ is a normalization factor. We show that a distribution trained with the proposed target converges to the distribution of $Z^\pi$ in Appendix A.1. If the target for time step $t+1$ is obtained, the target distribution for time step $t$ becomes the weighted sum of the current one-step TD target and the shifted previous target distribution, so it can be obtained recursively, as shown in (7). However, to obtain the target distribution, we need to store all quantile positions and weights for all time steps, which is not memory-efficient. Therefore, we propose to project the target distribution into a quantile distribution with a specific number of atoms, $M'$ (we set $M' > M$ to reduce information loss). The overall process to get the TD($\lambda$) target distribution is illustrated in Figure 1, and the pseudocode is given in Appendix A.2. After calculating the target distribution for all time steps, the critic can be trained to reduce the quantile regression loss with the target distribution.

## 3.2 SAFE DISTRIBUTIONAL ACTOR-CRITIC

**SAC-Style Surrogates.** Here, we derive efficient surrogate functions for the trust region method. While there are two main streams in trust region methods: trust region policy optimization (TRPO) (Schulman et al., 2015) and proximal policy optimization (PPO) (Schulman et al., 2017), however, we only consider TRPO since PPO is an approximation of TRPO by only considering the sum of rewards and, hence, cannot reflect safety constraints. There are several variants of TRPO (Nachum et al., 2018; Wang et al., 2017), among which off-policy TRPO (Meng et al., 2022) shows significantly improved sample efficiency by using off-policy data. Still, the performance of SAC outperforms off-policy TRPO (Meng et al., 2022), so we extend the surrogate of off-policy TRPO similar

to the policy loss of SAC. To this end, the surrogate should **1)** have entropy regularization and **2)** be expressed with Q-functions. If we define the objective function with entropy regularization as: $J(\pi) := \mathbb{E}\left[\sum_{t=0}^{\infty} \gamma^t (R(s_t, a_t, s_{t+1}) + \beta H(\pi(\cdot|s_t)))|\rho, \pi, P\right]$, where $H$ is the Shannon entropy, we can defined the following surrogate function:

$$J^{\mu,\pi}(\pi') := \mathbb{E}_{s_0 \sim \rho}[V^\pi(s_0)] + \frac{1}{1-\gamma}\left(\mathbb{E}_{d^\mu,\mu}\left[\frac{\pi'(a|s)}{\mu(a|s)}A^\pi(s,a)\right] + \beta\mathbb{E}_{d^\pi}[H(\pi'(\cdot|s))]\right), \quad (8)$$

where $\mu, \pi, \pi'$ are behavioral, current, and next policies, respectively. Then, we can derive a bound on the difference between the objective and surrogate functions.

**Theorem 1.** *Let us assume that* $\max_s H(\pi(\cdot|s)) < \infty$ *for* $\forall \pi \in \Pi$. *The difference between the objective and surrogate functions is bounded by a term consisting of KL divergence as:*

$$\left|J(\pi') - J^{\mu,\pi}(\pi')\right| \le \frac{\gamma}{(1-\gamma)^2}\sqrt{D_{\mathrm{KL}}^{\max}(\pi||\pi')}\left(\sqrt{2}\beta\epsilon_H + 2\epsilon_R\sqrt{D_{\mathrm{KL}}^{\max}(\mu||\pi')}\right), \quad (9)$$

*where* $\epsilon_H = \max_s |H(\pi'(\cdot|s))|$, $\epsilon_R = \max_{s,a}|A^\pi(s,a)|$, $D_{\mathrm{KL}}^{\max}(\pi||\pi') = \max_s D_{\mathrm{KL}}(\pi(\cdot|s)||\pi'(\cdot|s))$, *and the equality holds when* $\pi' = \pi$.

We provide the proof in Appendix A.3. Theorem 1 demonstrates that the surrogate function can approximate the objective function with a small error if the KL divergence is kept small enough. We then introduce a *SAC-style surrogate* by replacing the advantage in (8) with Q-function as follows:

$$J^{\mu,\pi}(\pi') = \frac{1}{1-\gamma}\left(\mathbb{E}_{d^\mu,\pi'}[Q^\pi(s,a)] + \beta\mathbb{E}_{d^\pi}[H(\pi'(\cdot|s))]\right) + C, \quad (10)$$

where $C$ is a constant term for $\pi'$. The policy gradient can be calculated using the reparameterization trick, as done in SAC (Haarnoja et al., 2018). We present the training results on continuous RL tasks in Appendix B, where the entropy-regularized (8) and SAC-style (10) versions are compared. Although (8) and (10) are mathematically equivalent, it can be observed that the performance of the SAC-style version is superior to the regularized version. We can analyze this with two factors. First, if using (8) in the off-policy setting, the importance ratios have significant variances, making training unstable. Second, the advantage function only gives scalar information about whether the sampled action is proper, whereas the Q-function directly gives the direction in which the action should be updated, so more information can be obtained from (10).

**Safe Policy Update.** Now, we can apply the same reformulation to the surrogate functions for the safety constraints, which are defined by Kim & Oh (2022a). The cost surrogate functions $F_k^{\mu,\pi}$ can be written in SAC-style form as follows:

$$J_{C_k}^{\mu,\pi}(\pi') := \mathbb{E}_{s\sim\rho}\left[V_{C_k}^\pi(s)\right] + \frac{1}{1-\gamma}\mathbb{E}_{d^\mu,\pi'}\left[Q_{C_k}^\pi(s,a)\right] - \frac{1}{1-\gamma}\mathbb{E}_{d^\mu}\left[V_{C_k}^\pi(s)\right],$$

$$J_{S_k}^{\mu,\pi}(\pi') := \mathbb{E}_{s\sim\rho}\left[S_{C_k}^\pi(s)\right] + \frac{1}{1-\gamma^2}\mathbb{E}_{d_2^\mu,\pi'}\left[S_{C_k}^\pi(s,a)\right] - \frac{1}{1-\gamma^2}\mathbb{E}_{d_2^\mu}\left[S_{C_k}^\pi(s)\right], \quad (11)$$

$$F_k^{\mu,\pi}(\pi';\alpha) := J_{C_k}^{\mu,\pi}(\pi') + \frac{\phi(\Phi^{-1}(\alpha))}{\alpha}\sqrt{J_{S_k}^{\mu,\pi}(\pi') - (J_{C_k}^{\mu,\pi}(\pi'))^2}.$$

Remark that Kim & Oh (2022a) have shown that the cost surrogates are bounded in terms of KL divergence between the current and next policy. Thus, we can construct the following practical, safe policy update rule by adding a trust region constraint:

$$\pi^{\mathrm{new}} = \underset{\pi'}{\arg\max}\, J^{\mu,\pi}(\pi') \text{ s.t. } F_k^{\mu,\pi}(\pi';\alpha) \le d_k \,\forall k = 1, ..., K, \, D_{\mathrm{KL}}(\pi||\pi') \le \epsilon, \quad (12)$$

where $D_{\mathrm{KL}}(\pi||\pi') := \mathbb{E}_{s\sim d_\mu}[D_{\mathrm{KL}}(\pi(\cdot|s)||\pi'(\cdot|s))]$, and $\epsilon$ is a trust region size. As (12) is non-linear, the objective and constraints are approximated linearly, while the KL divergence is approximated quadratically in order to determine the update direction. After the direction is obtained, a backtracking line search is performed. For more details, see Appendix A.5.

**Approximations.** In the distributional RL setting, the cost value and the cost square value functions can be approximated using the quantile distribution critics as follows:

$$Q_C^\pi(s,a) = \int_{-\infty}^{\infty} z\Pr(Z_C^\pi(s,a) = z)dz \approx \frac{1}{M}\sum_{m=1}^{M}\theta_m(s,a),$$

$$S_C^\pi(s,a) = \int_{-\infty}^{\infty} z^2\Pr(Z_C^\pi(s,a) = z)dz \approx \frac{1}{M}\sum_{m=1}^{M}\theta_m(s,a)^2. \quad (13)$$

Finally, the proposed method is summarized in Algorithm 1.

---

**Algorithm 1:** Safe Distributional Actor-Critic

---

**Data:** Policy network $\pi_\psi$, reward and cost critic networks $Z_{R,\theta}^\pi$, $Z_{C_k,\theta}^\pi$, and replay buffer $\mathcal{D}$.
Initialize network parameters $\psi, \theta$, and replay buffer $\mathcal{D}$.
**for** *epochs=1, E* **do**

    **for** *t=1, T* **do**

        Sample $a_t \sim \pi_\psi(\cdot|s_t)$ and get $s_{t+1}, r_t = R(s_t, a_t, s_{t+1}), c_{k,t} = C_k(s_t, a_t, s_{t+1})$.
        Store $(s_t, a_t, \pi_\psi(a_t|s_t), r_t, c_{k,t}, s_{t+1})$ in $\mathcal{D}$.

    **end**

    Calculate the TD($\lambda$) target distribution (Section 3.1) with $\mathcal{D}$ and update the critics to minimize (4).
    Calculate the surrogate (10) and the cost surrogates (11) with $\mathcal{D}$.
    Update the policy by solving (12), but if (12) has no solution, take a recovery step (Section 3.3).

**end**

---

### 3.3 FEASIBILITY HANDLING FOR MULTIPLE CONSTRAINTS

The proposed method updates a policy using (12), but the feasible set of (12) can be empty in the infeasible starting cases. To address the feasibility issue in safe RL with multiple constraints, one of the violated constraints can be selected, and the policy is updated to minimize the constraint until the feasible region is not empty (Xu et al., 2021), which is called a *naive approach*. However, it may not be easy to quickly reach the feasible condition if only one constraint at each update step is used to update the policy. Therefore, we propose a gradient integration method to reflect all the constraints simultaneously. The main idea is to get

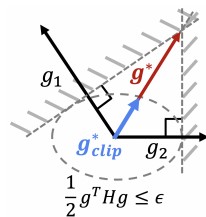

Figure 2: Gradient Integration.

a gradient that reduces the value of violated constraints and keeps unviolated constraints. To find such a gradient, the following quadratic program (QP) can be formulated by linearly approximating the constraints:

$$g^* = \underset{g}{\operatorname{argmin}} \frac{1}{2} g^T H g \ \text{ s.t. } g_k^T g + c_k \le 0, \ \forall k \in \{1, ..., K\}, \tag{14}$$

where $H$ is the Hessian of KL divergence at the current policy parameters $\psi$, $g_k$ is the gradient of the $k$th cost surrogate, $c_k = \min(\sqrt{2\epsilon g_k^T H^{-1} g_k}, F_k(\pi_\psi; \alpha) - d_k + \zeta)$, $\epsilon$ is a trust region size, and $\zeta \in \mathbb{R}_{>0}$ is a slack coefficient. Finally, we update the policy by $\psi^* = \psi + \min(1, \sqrt{2\epsilon/(g^{*T} H g^*)}) g^*$. Figure 2 illustrates the proposed gradient integration process. Each constraint is truncated by $c_k$ to be tangent to the trust region, and the slanted lines show the feasible region of truncated constraints. The solution of (14) is indicated in red, pointing to the nearest point in the intersection of the constraints. If the solution crosses the trust region, parameters are updated by the clipped direction, shown in blue. Then, the policy can reach the feasibility condition within finite time steps.

**Theorem 2.** *Assume that the cost surrogates are differentiable and convex, gradients of the surrogates are L-Lipschitz continuous, eigenvalues of the Hessian are equal or greater than a positive value $R \in \mathbb{R}_{>0}$, and $\{\psi|F_k(\pi_\psi; \alpha) + \zeta < d_k, \ \forall k\} \ne \emptyset$. Then, there exists $E \in \mathbb{R}_{>0}$ such that if $0 < \epsilon \le E$ and a policy is updated by the proposed gradient integration method, all constraints are satisfied within finite time steps.*

We provide the proof and show the existence of a solution (14) in Appendix A.4. The provided proof shows that the constant $E$ is proportional to $\zeta$ and inversely proportional to the number of constraints $K$. This means that the trust region size should be set smaller as $K$ increases and $\zeta$ decreases. In conclusion, if the policy update rule (12) is not feasible, a finite number of applications of the proposed gradient integration method will make the policy feasible.

## 4 RELATED WORK

**Safe Reinforcement Learning.** There are various safe RL methods depending on how to update policies to reflect safety constraints. First, trust region-based methods (Achiam et al., 2017; Yang et al., 2020; Kim & Oh, 2022a) find policy update directions by approximating the safe RL problem and update policies through a line search. Second, Lagrangian-based methods (Stooke et al., 2020;

Yang et al., 2021; Liu et al., 2020) convert the safe RL problem to a dual problem and update the policy and dual variables simultaneously. Last, expectation-maximization (EM) based methods (Liu et al., 2022) find non-parametric policy distributions by solving the safe RL problem in E-steps and fit parametric policies to the found non-parametric distributions in M-steps. Also, there are other ways to reflect safety other than policy updates. Qin et al. (2021); Lee et al. (2022) find optimal state or state-action distributions that satisfy constraints, and Bharadhwaj et al. (2021); Thananjeyan et al. (2021) reflect safety during exploration by executing only safe action candidates. In the experiments, only the safe RL methods of the policy update approach are compared with the proposed method.

**Distributional TD($\lambda$).** TD($\lambda$) (Precup et al., 2000) can be extended to the distributional critic to trade off bias-variance. Gruslys et al. (2018) have proposed a method to obtain target distributions by mixing $n$-step distributions, but the method is applicable only in discrete action spaces. Nam et al. (2021) have proposed a method to obtain target distributions using sampling to apply to continuous action spaces, but this is only for on-policy settings. A method proposed by Tang et al. (2022) updates the critics using newly defined distributional TD errors rather than target distributions. This method is applicable for off-policy settings but has the disadvantage that memory usage increases linearly with the number of TD error steps. In contrast to these methods, the proposed method is memory-efficient and applicable for continuous action spaces under off-policy settings.

**Gradient Integration.** The proposed feasibility handling method utilizes a gradient integration method, which is widely used in multi-task learning (MTL). The gradient integration method finds a single gradient to improve all tasks by using gradients of all tasks. Yu et al. (2020) have proposed a projection-based gradient integration method, which is guaranteed to converge Pareto-stationary sets. A method proposed by Liu et al. (2021) can reflect user preference, and Navon et al. (2022) proposed a gradient-scale invariant method to prevent the training process from being biased by a few tasks. The proposed method can be viewed as a mixture of projection and scale-invariant methods as gradients are clipped and projected onto a trust region.

## 5 EXPERIMENTS

We evaluate the safety performance of the proposed method and answer whether safe RL actually has the benefit of reducing the effort of reward engineering. For evaluation, agents are trained in the Safety Gym (Ray et al., 2019) with several tasks and robots. To check the advantage of safe RL, we construct locomotion tasks using legged robots with different models and different numbers of legs.

### 5.1 SAFETY GYM

**Tasks.** We employ two robots, point and car, to perform goal and button tasks in the Safety Gym. The goal task is to control a robot toward a randomly spawned goal without passing through hazard regions. The button task is to click a randomly designated button using a robot, where not only hazard regions but also dynamic obstacles exist. Agents get a cost when touching undesignated buttons and obstacles or entering hazard regions. There is only one constraint for all tasks, and it is defined using (3) with the sum of costs. Constraint violations (CVs) are counted when a robot contacts obstacles, unassigned buttons, or passes through hazard regions.

**Baselines.** Safe RL methods based on various types of policy updates are used as baselines. For the trust region-based method, we use constrained policy optimization (CPO) (Achiam et al., 2017) and off-policy trust-region CVaR (OffTRC) (Kim & Oh, 2022a), which extend the CPO to an off-policy and mean-std constrained version. For the Lagrangian-based method, worst-case soft actor-critic (WCSAC) (Yang et al., 2021) is used, and constrained variational policy optimization (CVPO) (Liu et al., 2022) based on the EM method is used. Specifically, WCSAC, OffTRC, and the proposed method, SDAC, use the mean-std constraints, so we experiment with those for $\alpha = 0.25, 0.5$, and $1.0$ (when $\alpha = 1.0$, the constraint is identical to the mean constraint).

**Results.** The graph of the final score and the total number of CVs are shown in Figure 3, and the training curves are provided in Appendix D.1. If points are located in the upper left corner of the graph, the result can be interpreted as excellent since the score is high and the number of CVs is low. The frontiers of SDAC, indicated by the blue dashed lines in Figure 3, are located in the upper left corners for all tasks. Hence, SDAC shows outstanding safety performance compared to other methods. In particular, SDAC with $\alpha = 0.25$ shows comparable scores despite recording the lowest

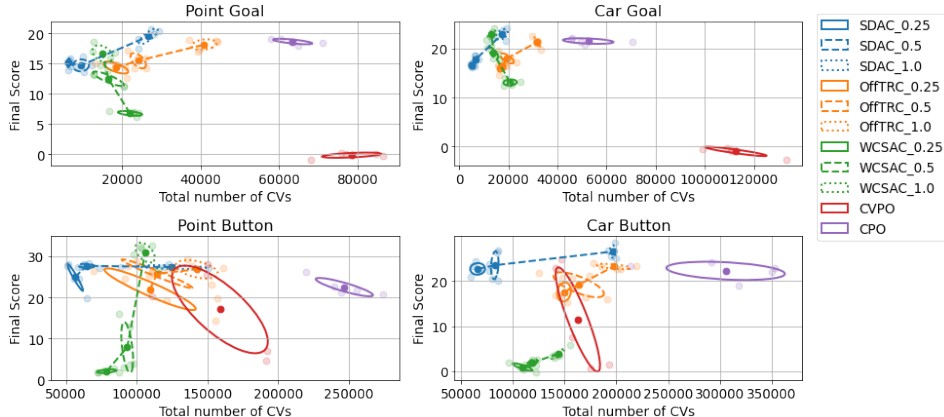

Figure 3: Graphs of final scores and the total number of CVs for the Safety Gym tasks. The number after the algorithm name in the legend indicates $\alpha$ used for the constraint. The center and boundary of ellipses are drawn using the mean and covariance of the five runs for each method. Dashed lines connect results of the same method with different $\alpha$.

number of CVs in all tasks. Although the frontier overlaps with WCSAC in the car goal and point button tasks, WCSAC shows a high fluctuation of scores depending on the value of $\alpha$. In addition, it can be seen that the proposed method enhances the efficiency and the training stability of the trust region method since SDAC shows high performance and small covariance compared to the other trust region-based methods, OffTRC and CPO.

## 5.2 LOCOMOTION TASKS

**Tasks.** The locomotion tasks are to train robots to follow $xy$-directional linear and $z$-directional angular velocity commands. Mini-Cheetah from MIT (Katz et al., 2019) and Laikago from Unitree (Wang, 2018) are used for quadrupedal robots, and Cassie from Agility Robotics (Xie et al., 2018) is used for a bipedal robot. In order to successfully perform the locomotion tasks, robots should keep balancing, standing, and stamping their feet so that they can move in any direction. Therefore, we define three constraints. The first constraint for balancing is to keep the body angle from deviating from zero, and the second for standing is to keep the height of the CoM above a threshold. The final constraint is to match the current foot contact state with a predefined foot contact timing. Especially, the contact timing is defined as stepping off the left and right feet symmetrically. The reward is defined as the negative $l^2$-norm of the difference between the command and the current velocity. For more details, see Appendix C.

**Baselines.** Through these tasks, we check the advantage of safe RL over traditional RL. Proximal policy optimization (PPO) (Schulman et al., 2017), based on the trust region method, and truncated quantile critic (TQC) (Kuznetsov et al., 2020), based on the SAC, are used as traditional RL baselines. To apply the same experiment to traditional RL, it is necessary to design a reward reflecting safety. We construct the reward through a weighted sum as $\bar{R} = (R - \sum_{i=1}^{3} w_i C_i)/(1 + \sum_{i=1}^{3} w_i)$, where $R$ and $C_{\{1,2,3\}}$ are used to train safe RL methods and are defined in Appendix C, and $R$ is called the *true reward*. The optimal weights are searched by a Bayesian optimization tool[1], which optimizes the true reward of PPO for the Mini-Cheetah task. The same weights are used for all robots and baselines to verify if reward engineering is required individually for each robot.

**Results.** Figure 4 shows the true reward sum graphs according to the $x$-directional velocity command. The overall training curves are presented in Appendix D.2, and the demonstration videos are attached to the supplementary. The figure shows that SDAC performs the locomotion tasks successfully, observing that the reward sums of all tasks are almost zero. PPO shows comparable results in the Mini-Cheetah and Laikago since the reward of the traditional RL baselines is optimized for the Mini-Cheetah task of PPO. However, the reward sum is significantly reduced in the Cassie task, where the kinematic model largely differs from the other robots. TQC shows the lowest reward sums

---

[1]We use Sweeps from Weights & Biases (Biewald, 2020).

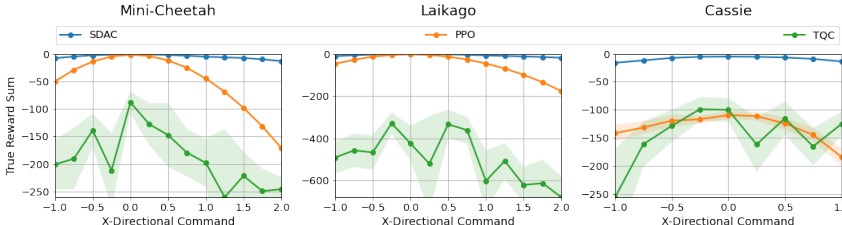

Figure 4: True reward sum graphs according to the x-directional command. The true reward is defined in Appendix C. The solid line and shaded area represent average and one fifth of std value, respectively. The graphs are obtained by running ten episodes per seed for each command.

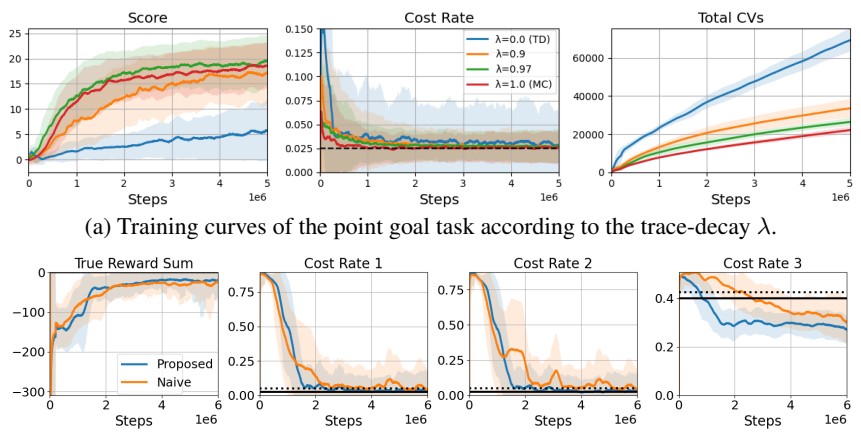

(a) Training curves of the point goal task according to the trace-decay $\lambda$.

(b) Training curves of the naive and proposed methods for the Cassie task.

Figure 5: Ablation results. The cost rates show the cost sums divided by the episode length. The shaded area represents the standard deviation. The black lines indicate the limit values, and the dotted lines in (b) represent the limit values $+ 0.025$.

despite the state-of-the-art algorithm in other RL benchmarks (Kuznetsov et al., 2020). From these results, it can be observed that reward engineering is required according to algorithms and robots.

## 5.3 ABLATION STUDY

We conduct ablation studies to show whether the proposed target distribution lowers the estimation bias and whether the proposed gradient integration quickly converges to the feasibility condition. In Figure 5a, the number of CVs is reduced as $\lambda$ increases, which means that the bias of constraint estimation decreases. However, the score also decreases due to large variance, showing that $\lambda$ can adjust the bias-variance tradeoff. In Figure 5b, the proposed gradient integration method is compared with a naive approach, which minimizes the constraints in order from the first to the third constraint, as described in Section 3.3. The proposed method reaches the feasibility condition faster than the naive approach and shows stable training curves because it reflects all constraints concurrently.

## 6 CONCLUSION

We have presented the trust region-based safe distributional RL method, called *SDAC*. To maximize the merit of the trust region method that can consistently satisfy constraints, we increase the performance by using the Q-function instead of the advantage function in the policy update. We have also proposed the memory-efficient, practical method for finding low-biased target distributions in off-policy settings to estimate constraints. Finally, we proposed the handling method for multiple constraint settings to solve the feasibility issue caused when using the trust region method. From extensive the experiments, we have demonstrated that SDAC with mean-std constraints achieved improved performance with minimal constraint violations and successfully performed the locomotion tasks without reward engineering.

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

# A ALGORITHM DETAILS

## A.1 CONVERGENCE ANALYSIS

In this section, we show that the proposed TD($\lambda$) target distribution converges to the $Z^\pi$. First, we express the target distribution using a distributional operator and show that the operator is contractive. Finally, we show that $Z^\pi$ is the unique fixed point.

Before starting the proof, we introduce useful notions, distance metrics, and operators. As the return $Z^\pi(s, a)$ is a random variable, we define the distribution of $Z^\pi(s, a)$ as $\nu^\pi(s, a)$. Let $\eta$ be the distribution of a random variable $X$. Then, we can express the distribution of affine transformation of random variable, $aX + b$, using the *pushforward* operator, which is defined by Rowland et al. (2018), as $(f_{a,b})_\#(\eta)$. To measure a distance between two distributions, Bellemare et al. (2023) has defined the distance $l_p$ as follows:

$$l_p(\eta_1, \eta_2) := \left( \int_\mathbb{R} |F_{\eta_1}(x) - F_{\eta_2}(x)|^p \, dx \right)^{1/p}, \tag{15}$$

where $F_\nu(z)$ is the cumulative distribution function. This distance is $1/p$-homogeneous, regular, and $p$-convex (see Section 4 of Bellemare et al. (2023) for more details). For functions that map state-action pairs to distributions, a distance can be defined as (Bellemare et al., 2023): $\bar{l}_p(\nu_1, \nu_2) := \sup_{(s,a) \in S \times A} l_p(\nu_1(s, a), \nu_2(s, a))$. Then, the proposed TD($\lambda$) target distribution can be expressed as an operator as below.

$$\mathcal{T}_\lambda^{\mu,\pi} \nu(s, a) := \frac{1-\lambda}{\mathcal{N}} \sum_{i=0}^\infty \lambda^i$$
$$\times \mathbb{E}_\mu \left[ \left( \prod_{j=1}^i \eta(s_j, a_j) \right) \mathbb{E}_{a' \sim \pi(\cdot|s_{i+1})} \left[ (f_{\gamma^{i+1}, \sum_{t=0}^i \gamma^t r_t})_\#(\nu(s_{i+1}, a')) \right] \Big| s_0 = s, a_0 = a \right], \tag{16}$$

where $\eta(s, a) = \frac{\pi(a|s)}{\mu(a|s)}$. Then, the operator $\mathcal{T}_\lambda^{\mu,\pi}$ has a contraction property.

**Theorem 3.** *Under the distance $\bar{l}_p$ and the assumption that the state, action, and reward spaces are finite, $\mathcal{T}_\lambda^{\mu,\pi}$ is $\gamma^{1/p}$-contractive.*

*Proof.* First, the operator can be rewritten using summation as follows.

$$\mathcal{T}_\lambda^{\mu,\pi} \nu(s, a) = \frac{1-\lambda}{\mathcal{N}} \sum_{i=0}^\infty \lambda^i \sum_{a' \in A} \sum_{(s_0, a_0, r_0, \ldots, s_{i+1})} \Pr\!_\mu(\underbrace{s_0, a_0, r_0, \ldots, s_{i+1}}_{=:\tau}) \left( \prod_{j=1}^i \eta(s_j, a_j) \right)$$
$$\times \pi(a'|s_{i+1})(f_{\gamma^{i+1}, \sum_{t=0}^i \gamma^t r_t})_\#(\nu(s_{i+1}, a'))$$
$$= \frac{1-\lambda}{\mathcal{N}} \sum_{i=0}^\infty \lambda^i \sum_{a' \in A} \sum_\tau \Pr\!_\mu(\tau) \left( \prod_{j=1}^i \eta(s_j, a_j) \right) \pi(a'|s_{i+1}) \sum_{s' \in S} \mathbf{1}_{s'=s_{i+1}}$$
$$\times \sum_{r'_{0:i}} \left( \prod_{k=0}^i \mathbf{1}_{r'_k=r_k} \right) (f_{\gamma^{i+1}, \sum_{t=0}^i \gamma^t r'_t})_\#(\nu(s', a'))$$
$$= \frac{1-\lambda}{\mathcal{N}} \sum_{i=0}^\infty \lambda^i \sum_{a' \in A} \sum_{s' \in S} \sum_{r'_{0:i}} (f_{\gamma^{i+1}, \sum_{t=0}^i \gamma^t r'_t})_\#(\nu(s', a'))$$
$$\times \underbrace{\mathbb{E}_\mu \left[ \left( \prod_{j=1}^i \eta(s_j, a_j) \right) \pi(a'|s_{i+1}) \mathbf{1}_{s'=s_{i+1}} \left( \prod_{k=0}^i \mathbf{1}_{r'_k=r_k} \right) \right]}_{=:w_{s',a',r'_{0:i}}} \tag{17}$$
$$= \frac{1-\lambda}{\mathcal{N}} \sum_{i=0}^\infty \sum_{s' \in S} \sum_{a' \in A} \sum_{r'_{0:i}} \lambda^i w_{s',a',r'_{0:i}} (f_{\gamma^{i+1}, \sum_{t=0}^i \gamma^t r'_t})_\#(\nu(s', a')).$$

Since the sum of weights of distributions should be one, we can find the normalization factor $\mathcal{N} = (1-\lambda)\sum_{i=0}^{\infty}\sum_{s\in S}\sum_{a\in A}\sum_{r_{0:i}}\lambda^i w_{s,a,r_{0:i}}$. Then, the following inequality can be derived using the homogeneity, regularity, and convexity of $l_p$:

$$
\begin{aligned}
& l_p^p(\mathcal{T}_\lambda^{\mu,\pi}\nu_1(s,a), \mathcal{T}_\lambda^{\mu,\pi}\nu_2(s,a)) \\
&= l_p^p\left(\frac{1-\lambda}{\mathcal{N}}\sum_{i=0}^{\infty}\sum_{s\in S}\sum_{a\in A}\sum_{r_{0:i}}\lambda^i w_{s,a,r_{0:i}}(f_{\gamma^{i+1},\sum_{t=0}^{i}\gamma^t r_t})\#(\nu_1(s,a)),\right. \\
&\qquad\qquad \left.\frac{1-\lambda}{\mathcal{N}}\sum_{i=0}^{\infty}\sum_{s\in S}\sum_{a\in A}\sum_{r_{0:i}}\lambda^i w_{s,a,r_{0:i}}(f_{\gamma^{i+1},\sum_{t=0}^{i}\gamma^t r_t})\#(\nu_2(s,a))\right) \\
&\leq \sum_{i=0}^{\infty}\sum_{s\in S}\sum_{a\in A}\sum_{r_{0:i}}\frac{(1-\lambda)\lambda^i w_{s,a,r_{0:i}}}{\mathcal{N}}l_p^p\left((f_{\gamma^{i+1},\sum_{t=0}^{i}\gamma^t r_t})\#(\nu_1(s,a)),\right. \\
&\qquad\qquad\qquad\qquad\qquad \left.(f_{\gamma^{i+1},\sum_{t=0}^{i}\gamma^t r_t})\#(\nu_2(s,a))\right) \qquad\qquad (18) \\
&\leq \sum_{i=0}^{\infty}\sum_{s\in S}\sum_{a\in A}\sum_{r_{0:i}}\frac{(1-\lambda)\lambda^i w_{s,a,r_{0:i}}}{\mathcal{N}}l_p^p\left((f_{\gamma^{i+1},0})\#(\nu_1(s,a)),(f_{\gamma^{i+1},0})\#(\nu_2(s,a))\right) \\
&= \sum_{i=0}^{\infty}\sum_{s\in S}\sum_{a\in A}\sum_{r_{0:i}}\frac{(1-\lambda)\lambda^i w_{s,a,r_{0:i}}}{\mathcal{N}}\gamma^{i+1}l_p^p\left(\nu_1(s,a),\nu_2(s,a)\right) \\
&\leq \sum_{i=0}^{\infty}\sum_{s\in S}\sum_{a\in A}\sum_{r_{0:i}}\frac{(1-\lambda)\lambda^i w_{s,a,r_{0:i}}}{\mathcal{N}}\gamma^{i+1}\left(\bar{l}_p\left(\nu_1,\nu_2\right)\right)^p \\
&\leq \gamma\left(\bar{l}_p\left(\nu_1,\nu_2\right)\right)^p.
\end{aligned}
$$

Therefore, $\bar{l}_p\left(\mathcal{T}_\lambda^{\mu,\pi}\nu_1,\mathcal{T}_\lambda^{\mu,\pi}\nu_2\right)\leq \gamma^{1/p}\bar{l}_p\left(\nu_1,\nu_2\right)$. $\qquad\qquad\qquad\qquad\square$

By the Banach's fixed point theorem, the operator has a unique fixed distribution. From the definition of $Z^\pi$, the following equality holds (Rowland et al., 2018): $\nu^\pi(s,a) = \mathbb{E}_\pi\left[(f_{\gamma,r})\#(\nu^\pi(s',a'))\right]$. Then, it can be shown that $\nu^\pi$ is the fixed distribution by applying the operator $\mathcal{T}_\lambda^{\mu,\pi}$ to $\nu^\pi$:

$$
\begin{aligned}
\mathcal{T}_\lambda^{\mu,\pi}\nu^\pi(s,a) &= \frac{1-\lambda}{\mathcal{N}}\sum_{i=0}^{\infty}\lambda^i \\
&\times \mathbb{E}_\mu\left[\left(\prod_{j=1}^{i}\eta(s_j,a_j)\right)\mathbb{E}_{a'\sim\pi(\cdot|s_{i+1})}\left[(f_{\gamma^{i+1},\sum_{t=0}^{i}\gamma^t r_t})\#(\nu^\pi(s_{i+1},a'))\right]\Big| s_0=s,a_0=a\right] \\
&= \frac{1-\lambda}{\mathcal{N}}\sum_{i=0}^{\infty}\lambda^i\mathbb{E}_\pi\left[(f_{\gamma^{i+1},\sum_{t=0}^{i}\gamma^t r_t})\#(\nu^\pi(s_{i+1},a_{i+1}))\Big| s_0=s,a_0=a\right] \\
&= \frac{1-\lambda}{\mathcal{N}}\sum_{i=0}^{\infty}\lambda^i\nu^\pi(s,a) = \nu^\pi(s,a).
\end{aligned}
\qquad (19)
$$

## A.2 Pseudocode of TD($\lambda$) Target Distribution

We provide the pseudocode for calculating TD($\lambda$) target distribution for the reward critic in Algorithm 2. The target distribution for the cost critics can also be obtained by simply replacing the reward part with the cost.

---

**Algorithm 2:** TD($\lambda$) Target Distribution

---

**Data:** Policy network $\pi_\psi$, critic network $Z_\theta^\pi$, and trajectory $\{(s_t, a_t, \mu(a_t|s_t), r_t, d_t, s_{t+1})\}_{t=1}^T$.
Sample an action $a'_{T+1} \sim \pi_\psi(s_{T+1})$ and get $\hat{Z}_T^{\text{tot}} = r_T + (1 - d_T)\gamma Z_\theta^\pi(s_{T+1}, a'_{T+1})$.
Initialize the total weight $w_{\text{tot}} = \lambda$.
**for** *t=T, 1* **do**

    Sample an action $a'_{t+1} \sim \pi_\psi(s_{t+1})$ and get $\hat{Z}_t^{(1)} = r_t + (1 - d_t)\gamma Z_\theta^\pi(s_{t+1}, a'_{t+1})$.
    Set the current weight $w = 1 - \lambda$.
    Combine the two targets, $(\hat{Z}_t^{(1)}, w)$ and $(\hat{Z}_t^{(\text{tot})}, w_{\text{tot}})$, and sort the combined target
      according to the positions of atoms.
    Build the CDF of the combined target by accumulating the weights at each atom.
    Project the combined target into a quantile distribution with $M'$ atoms, which is $\hat{Z}_t^{(\text{proj})}$,
      using the CDF (find the atom positions corresponding to each quantile).
    Update $\hat{Z}_{t-1}^{(\text{tot})} = r_{t-1} + (1 - d_{t-1})\gamma\hat{Z}_t^{(\text{proj})}$ and
    $w_{\text{tot}} = \lambda\frac{\pi_\psi(a_t|s_t)}{\mu(a_t|s_t)}(1 - d_{t-1})(1 - \lambda + w_{\text{tot}})$.

**end**
**Return** $\{\hat{Z}_t^{(\text{proj})}\}_{t=1}^T$.

---

## A.3 Proof of Theorem 1

Before showing the proof, we present a new function and a lemma. A value difference function is defined as follows:

$$\delta^{\pi'}(s) := \mathbb{E}\left[R(s, a, s') + \gamma V^\pi(s') - V^\pi(s) \mid a \sim \pi'(\cdot|s), s' \sim P(\cdot|s, a)\right] = \mathop{\mathbb{E}}_{a \sim \pi'}\left[A^\pi(s, a)\right].$$

**Lemma 4.** *The maximum of $|\delta^{\pi'}(s) - \delta^\pi(s)|$ is equal or less than $\epsilon_R\sqrt{2D_{\text{KL}}^{\max}(\pi||\pi')}$.*

*Proof.* The value difference can be expressed in a vector form,

$$\delta^{\pi'}(s) - \delta^\pi(s) = \sum_a (\pi'(a|s) - \pi(a|s))A^\pi(s, a) = \langle \pi'(\cdot|s) - \pi(\cdot|s), A^\pi(s, \cdot) \rangle.$$

Using Hölder's inequality, the following inequality holds:

$$|\delta^{\pi'}(s) - \delta^\pi(s)| \leq ||\pi'(\cdot|s) - \pi(\cdot|s)||_1 \cdot ||A^\pi(s, \cdot)||_\infty$$
$$= 2D_{\text{TV}}(\pi'(\cdot|s)||\pi(\cdot|s))\max_a A^\pi(s, a).$$

$$\Rightarrow ||\delta^{\pi'} - \delta^\pi||_\infty = \max_s|\delta^{\pi'}(s) - \delta^\pi(s)| \leq 2\epsilon_R\max_s D_{\text{TV}}(\pi(\cdot|s)||\pi'(\cdot|s)).$$

Using Pinsker's inequality, $||\delta^{\pi'} - \delta^\pi||_\infty \leq \epsilon_R\sqrt{2D_{\text{KL}}^{\max}(\pi||\pi')}$. $\qquad\square$

**Theorem 1.** *Let us assume that $\max_s H(\pi(\cdot|s)) < \infty$ for $\forall\pi \in \Pi$. The difference between the objective and surrogate functions is bounded by a term consisting of KL divergence as:*

$$\left|J(\pi') - J^{\mu,\pi}(\pi')\right| \leq \frac{\gamma}{(1-\gamma)^2}\sqrt{D_{\text{KL}}^{\max}(\pi||\pi')}\left(\sqrt{2}\beta\epsilon_H + 2\epsilon_R\sqrt{D_{\text{KL}}^{\max}(\mu||\pi')}\right), \qquad (9)$$

*where $\epsilon_H = \max_s|H(\pi'(\cdot|s))|$, $\epsilon_R = \max_{s,a}|A^\pi(s, a)|$, $D_{\text{KL}}^{\max}(\pi||\pi') = \max_s D_{\text{KL}}(\pi(\cdot|s)||\pi'(\cdot|s))$, and the equality holds when $\pi' = \pi$.*

*Proof.* The surrogate function can be expressed in vector form as follows:

$$J^{\mu,\pi}(\pi') = \langle \rho, V^\pi \rangle + \frac{1}{1-\gamma} \left( \langle d^\mu, \delta^{\pi'} \rangle + \beta \langle d^\pi, H^{\pi'} \rangle \right),$$

where $H^{\pi'}(s) = H(\pi'(\cdot|s))$. The objective function of $\pi'$ can also be expressed in a vector form using Lemma 1 from Achiam et al. (2017),

$$\begin{aligned}
J(\pi') &= \frac{1}{1-\gamma} \mathbb{E} \left[ R(s,a,s') + \beta H^{\pi'}(s) \mid s \sim d^{\pi'}, a \sim \pi'(\cdot|s), s' \sim P(\cdot|s,a) \right] \\
&= \frac{1}{1-\gamma} \underset{s \sim d^{\pi'}}{\mathbb{E}} \left[ \delta^{\pi'}(s) + \beta H^{\pi'}(s) \right] + \underset{s \sim \rho}{\mathbb{E}} \left[ V^\pi(s) \right] \\
&= \langle \rho, V^\pi \rangle + \frac{1}{1-\gamma} \langle d^{\pi'}, \delta^{\pi'} + \beta H^{\pi'} \rangle.
\end{aligned}$$

By Lemma 3 from Achiam et al. (2017), $||d^\pi - d^{\pi'}||_1 \leq \frac{\gamma}{1-\gamma} \sqrt{2 D_{\mathrm{KL}}^{\max}(\pi||\pi')}$. Then, the following inequality is satisfied:

$$\begin{aligned}
|(1-\gamma)&(J^{\mu,\pi}(\pi') - J(\pi'))| \\
&= |\langle d^{\pi'} - d^\mu, \delta^{\pi'} \rangle + \beta \langle d^\pi - d^{\pi'}, H^{\pi'} \rangle| \\
&\leq |\langle d^{\pi'} - d^\mu, \delta^{\pi'} \rangle| + \beta |\langle d^\pi - d^{\pi'}, H^{\pi'} \rangle| \\
&= |\langle d^{\pi'} - d^\mu, \delta^{\pi'} - \delta^\pi \rangle| + \beta |\langle d^\pi - d^{\pi'}, H^{\pi'} \rangle| & (\because \delta^\pi = 0) \\
&\leq ||d^{\pi'} - d^\mu||_1 ||\delta^{\pi'} - \delta^\pi||_\infty + \beta ||d^\pi - d^{\pi'}||_1 ||H^{\pi'}||_\infty & (\because \text{Hölder's inequality}) \\
&\leq \frac{2\epsilon_R \gamma}{1-\gamma} \sqrt{D_{\mathrm{KL}}^{\max}(\mu||\pi') D_{\mathrm{KL}}^{\max}(\pi||\pi')} + \frac{\beta\gamma\epsilon_H}{1-\gamma} \sqrt{2 D_{\mathrm{KL}}^{\max}(\pi||\pi')} & (\because \text{Lemma 4}) \\
&= \frac{\gamma}{1-\gamma} \sqrt{D_{\mathrm{KL}}^{\max}(\pi||\pi')} \left( \sqrt{2}\beta\epsilon_H + 2\epsilon_R \sqrt{D_{\mathrm{KL}}^{\max}(\mu||\pi')} \right).
\end{aligned}$$

If $\pi' = \pi$, the KL divergence term becomes zero, so equality holds. $\qquad\square$

### A.4 PROOF OF THEOREM 2

We denote the policy parameter space as $\Psi \subseteq \mathbb{R}^d$, the parameter at the $t$th iteration as $\psi_t \in \Psi$, the Hessian matrix as $H(\psi_t) = \nabla_\psi^2 D_{\mathrm{KL}}(\pi_{\psi_t}||\pi_\psi)|_{\psi=\psi_t}$, and the $k$th cost surrogate as $F_k(\psi_t) = F_k^{\mu,\pi}(\pi_{\psi_t}; \alpha)$. As we focus on the $t$th iteration, the following notations are used for brevity: $H = H(\psi_t)$ and $g_k = \nabla F_k(\psi_t)$. The proposed gradient integration at $t$th iteration is defined as the following quadratic program (QP):

$$g_t = \underset{g}{\arg\min} \frac{1}{2} g^T H g \quad \text{s.t. } g_k^T g + c_k \leq 0 \text{ for } \forall k, \tag{20}$$

where $c_k = \min(\sqrt{2\epsilon g_k^T H^{-1} g_k}, F_k(\pi_\psi; \alpha) - d_k + \zeta)$. In the remainder of this section, we introduce the assumptions and new definitions, discuss the existence of a solution (20), show the convergence to the feasibility condition for varying step size cases, and provide the proof of Theorem 2.

**Assumption.** **1)** Each $F_k$ is differentiable and convex, **2)** $\nabla F_k$ is $L$-Lipschitz continuous, **3)** all eigenvalues of the Hessian matrix $H(\psi)$ are equal or greater than $R \in \mathbb{R}_{>0}$ for $\forall \psi \in \Psi$, and **4)** $\{\psi | F_k(\psi) + \zeta < d_k \text{ for } \forall k\} \neq \emptyset$.

**Definition.** Using the Cholesky decomposition, the Hessian matrix can be expressed as $H = B \cdot B^T$ where $B$ is a lower triangular matrix. By introducing new terms, $\bar{g}_k := B^{-1} g_k$ and $b_t := B^T g_t$, the following is satisfied: $g_k^T H^{-1} g_k = ||\bar{g}_k||_2^2$. Additionally, we define the in-boundary and out-boundary sets as:

$$\mathrm{IB}_k := \left\{ \psi | F_k(\psi) - d_k + \zeta \leq \sqrt{2\epsilon \nabla F_k(\psi)^T H^{-1}(\psi) \nabla F_k(\psi)} \right\},$$

$$\mathrm{OB}_k := \left\{ \psi | F_k(\psi) - d_k + \zeta \geq \sqrt{2\epsilon \nabla F_k(\psi)^T H^{-1}(\psi) \nabla F_k(\psi)} \right\}.$$

The minimum of $||\bar{g}_k||$ in $\text{OB}_k$ is denoted as $m_k$, and the maximum of $||\bar{g}_k||$ in $\text{IB}_k$ is denoted as $M_k$. Also, $\min_k m_k$ and $\max_k M_k$ is denoted as $m$ and $M$, respectively, and we can say that $m$ is positive.

**Lemma 5.** *For all $k$, the minimum value of $m_k$ is positive.*

*Proof.* Assume that there exist $k \in \{1, ..., K\}$ such that $m_k$ is equal to zero at a policy parameter $\psi^* \in \text{OB}_k$, i.e., $||\nabla F_k(\psi^*)|| = 0$. Since $F_k$ is convex, $\psi^*$ is a minimum point of $F_k$, $\min_\psi F_k(\psi) = F_k(\psi^*) < d_k - \zeta$. However, $F_k(\psi^*) \geq d_k - \zeta$ as $\psi^* \in \text{OB}_k$, so $m_k$ is positive due to the contradiction. Hence, the minimum of $m_k$ is also positive. $\square$

**Lemma 6.** *A solution of (20) always exists.*

*Proof.* There exists a policy parameter $\hat{\psi} \in \{\psi | F_k(\psi) + \zeta < d_k \text{ for } \forall k\}$ due to the assumptions. Let $g = \psi - \psi_t$. Then, the following inequality holds.

$$g_k^T(\psi - \psi_t) + c_k \leq g_k^T(\psi - \psi_t) + F_k(\psi_t) + \zeta - d_k \leq F_k(\psi) + \zeta - d_k. \quad (\because F_k \text{ is convex.})$$
$$\Rightarrow g_k^T(\hat{\psi} - \psi_t) + c_k \leq F_k(\hat{\psi}) + \zeta - d_k < 0 \text{ for } \forall k.$$

Since $\hat{\psi} - \psi_t$ satisfies all constraints of (20), the feasible set is non-empty and convex. Also, $H$ is positive definite, so the QP has a unique solution. $\square$

Lemma 6 shows the existence of solution of (20). Now, we show the convergence of the proposed gradient integration method in the case of varying step sizes.

**Lemma 7.** *If $\sqrt{2\epsilon}M \leq \zeta$ and a policy is updated by $\psi_{t+1} = \psi_t + \beta_t g_t$, where $0 < \beta_t < \frac{2\sqrt{2\epsilon}mR}{L||b_t||^2}$ and $\beta_t \leq 1$, the policy satisfies $F_k(\psi) \leq d_k$ for $\forall k$ within a finite time.*

*Proof.* We can reformulate the step size as $\beta = \frac{2\sqrt{2\epsilon}mR}{L||b_t||^2}\beta_t'$, where $\beta_t' \leq \frac{L||b_t||^2}{2\sqrt{2\epsilon}mR}$ and $0 < \beta_t' < 1$. Since the eigenvalues of $H$ is equal to or bigger than $R$ and $H$ is symmetric and positive definite, $\frac{1}{R}I - H^{-1}$ is positive semi-definite. Hence, $x^T H^{-1} x \leq \frac{1}{R}||x||^2$ is satisfied. Using this fact, the following inequality holds:

$$F_k(\psi_t + \beta_t g_t) - F_k(\psi_t) \leq \beta_t \nabla F_k(\psi_t)^T g_t + \frac{L}{2}||\beta_t g_t||^2 \quad (\because \nabla F_k \text{ is } L\text{-Lipschitz continuous.})$$
$$= \beta_t g_k^T g_t + \frac{L}{2}\beta_t^2 ||g_t||^2$$
$$= \beta_t g_k^T g_t + \frac{L}{2}\beta_t^2 b_t^T H^{-1} b_t \quad (\because g_t = B^{-T} b_t)$$
$$\leq -\beta_t c_k + \frac{L}{2R}\beta_t^2 ||b_t||^2. \quad (\because g_k^T g_t + c_k \leq 0)$$

Now, we will show that $\psi$ enters $\text{IB}_k$ in a finite time for $\forall \psi \in \text{OB}_k$ and that the $k$th constraint is satisfied for $\forall \psi \in \text{IB}_k$. Thus, we divide into two cases, **1)** $\psi_t \in \text{OB}_k$ and **2)** $\psi_t \in \text{IB}_k$. For the first case, $c_k = \sqrt{2\epsilon}||\bar{g}_k||$, so the following inequality holds:

$$F_k(\psi_t + \beta_t g_t) - F_k(\psi_t) \leq \beta_t \left( -\sqrt{2\epsilon}||\bar{g}_k|| + \frac{L}{2R}\beta_t ||b_t||^2 \right)$$
$$\leq \beta_t \sqrt{2\epsilon} \left( -||\bar{g}_k|| + m\beta_t' \right) \tag{21}$$
$$\leq \beta_t \sqrt{2\epsilon}m(\beta_t' - 1) < 0.$$

The value of $F_k$ decreases strictly with each update step according to (21). Hence, $\psi_t$ can reach $\text{IB}_k$ by repeatedly updating the policy. We now check whether the constraint is satisfied for the second case. For the second case, the following inequality holds by applying $c_k = F_k(\psi_t) - d_k + \zeta$:

$$F_k(\psi_t + \beta_t g_t) - F_k(\psi_t) \leq \beta_t d_k - \beta_t F_k(\psi_t) - \beta_t \zeta + \frac{L}{2R}\beta_t^2 ||b_t||^2$$
$$\Rightarrow F_k(\psi_t + \beta_t g_t) - d_k \leq (1 - \beta_t)(F_k(\psi_t) - d_k) + \beta_t(-\zeta + \sqrt{2\epsilon}m\beta_t').$$

Since $\psi_t \in \mathrm{IB}_k$,

$$F_k(\psi_t) - d_k \leq \sqrt{2\epsilon}||\bar{g}_k|| - \zeta \leq \sqrt{2\epsilon}M - \zeta \leq 0.$$

Since $m \leq M$ and $\beta'_t < 1$,

$$-\zeta + \sqrt{2\epsilon}m\beta'_t < -\zeta + \sqrt{2\epsilon}M \leq 0.$$

Hence, $F_k(\psi_t + \beta_t g_t) \leq d_k$, which means that the $k$th constraint is satisfied if $\psi_t \in \mathrm{IB}_k$. As $\psi_t$ reaches $\mathrm{IB}_k$ for $\forall k$ within a finite time according to (21), the policy can satisfy all constraints within a finite time. $\square$

Lemma 7 shows the convergence to the feasibility condition in the case of varying step sizes. We introduce a lemma, which shows $||b_t||$ is bounded by $\sqrt{\epsilon}$, and finally show the proof of Theorem 2, which can be considered a special case of varying step sizes.

**Lemma 8.** *There exists $T \in \mathbb{R}_{>0}$ such that $||b_t|| \leq T\sqrt{\epsilon}$.*

*Proof.* By solving the dual problem of (20), $g_t$ can be expressed as:

$$g_t = -\sum_{k=1}^{K} \lambda_k H^{-1} g_k \text{ s.t. } \lambda_k = \max\left(\frac{c_k - \sum_{j \neq k} \lambda_j g_j^T H^{-1} g_k}{g_k^T H^{-1} g_k}, 0\right) \text{ for } \forall k.$$

The following inequality holds for $\forall k$:

$$\lambda_k \leq \max\left(\frac{c_k}{||\bar{g}_k||^2}, 0\right) \leq \max\left(\frac{\sqrt{2\epsilon}||\bar{g}_k||}{||\bar{g}_k||^2}, 0\right) \leq \frac{\sqrt{2\epsilon}}{||\bar{g}_k||}.$$

Using triangular inequality,

$$||b_t|| = ||B^T g_t|| = ||\sum_k \lambda_k B^T H^{-1} g_k|| \leq \sum_k \lambda_k ||B^T H^{-1} g_k||$$

$$\leq \sqrt{2\epsilon} \sum_k \frac{||B^T H^{-1} g_k||}{||\bar{g}_k||} = K\sqrt{2\epsilon}.$$

Hence, for every constant $T > \sqrt{2}K$, the statement holds. $\square$

**Theorem 2.** *Assume that the cost surrogates are differentiable and convex, gradients of the surrogates are L-Lipschitz continuous, eigenvalues of the Hessian are equal or greater than a positive value $R \in \mathbb{R}_{>0}$, and $\{\psi|F_k(\pi_\psi; \alpha) + \zeta < d_k, \ \forall k\} \neq \emptyset$. Then, there exists $E \in \mathbb{R}_{>0}$ such that if $0 < \epsilon \leq E$ and a policy is updated by the proposed gradient integration method, all constraints are satisfied within finite time steps.*

*Proof.* The proposed step size is $\beta_t = \min(1, \sqrt{2\epsilon}/||b_t||)$, and the sufficient conditions that guarantee the convergence according to Lemma 7 are followings:

$$\sqrt{2\epsilon}M \leq \zeta \text{ and } 0 < \beta_t \leq 1 \text{ and } \beta_t < \frac{2\sqrt{2\epsilon}mR}{L||b_t||^2}.$$

The second condition is self-evident. To satisfy the third condition, the proposed step size $\beta_t$ should satisfy the followings:

$$\frac{\sqrt{2\epsilon}}{||b_t||} < \frac{2\sqrt{2\epsilon}mR}{L||b_t||^2} \Leftrightarrow ||b_t|| < \frac{2mR}{L}.$$

If $\epsilon < 2((mR)/(LK))^2$, the following inequality holds:

$$\sqrt{2\epsilon} < \frac{2mR}{LK} \Rightarrow ||b_t|| \leq K\sqrt{2\epsilon} < \frac{2mR}{L}. \quad (\because \text{ Lemma 8.})$$

Hence, if $\epsilon \leq E = \frac{1}{2}\min(\frac{\zeta^2}{2M^2}, 2(\frac{mR}{LK})^2)$, the sufficient conditions are satisfied. $\square$

## A.5 POLICY UPDATE RULE

To solve the constrained optimization problem (12), we find a policy update direction by linearly approximating the objective and safety constraints and quadratically approximating the trust region constraint, as done by Achiam et al. (2017). After finding the direction, we update the policy using a line search method. Given the current policy parameter $\psi_t \in \Psi$, the approximated problem can be expressed as follows:

$$x^* = \operatorname*{argmax}_{x \in \Psi} g^T x \quad \text{s.t.} \ \frac{1}{2} x^T H x \leq \epsilon, \ b_k^T x + c_k \leq 0 \ \forall k, \tag{22}$$

where $g = \nabla_\psi J^{\mu,\pi}(\pi_\psi)|_{\psi=\psi_t}$, $H = \nabla_\psi^2 D_{\mathrm{KL}}(\pi_{\psi_t}||\pi_\psi)|_{\psi=\psi_t}$, $b_k = \nabla_\psi F_k^{\mu,\pi}(\pi_\psi;\alpha)|_{\psi=\psi_t}$, and $c_k = F_k(\pi_\psi;\alpha) - d_k$. Since (22) is convex, we can use an existing convex optimization solver. However, the search space, which is the policy parameter space $\Psi$, is excessively large, so we reduce the space by converting (22) to a dual problem as follows:

$$g(\lambda, \nu) = \min_x L(x, \lambda, \nu) = \min_x \{ g^T x + \nu(\frac{1}{2} x^T H x - \epsilon) + \lambda^T (Bx + c) \}$$

$$= \frac{-1}{2\nu} \left( \underbrace{g^T H^{-1} g}_{=:q} + 2 \underbrace{g^T H^{-1} B^T}_{=:r^T} \lambda + \lambda^T \underbrace{B H^{-1} B^T}_{=:S} \lambda \right) + \lambda^T c - \nu \epsilon \tag{23}$$

$$= \frac{-1}{2\nu}(q + 2r^T \lambda + \lambda^T S \lambda) + \lambda^T c - \nu\epsilon,$$

where $B = (b_1, .., b_K)$, $c = (c_1, ..., c_K)^T$, and $\lambda \in \mathbb{R}^K \geq 0$ and $\nu \in \mathbb{R} \geq 0$ are Lagrange multipliers. Then, the optimal $\lambda$ and $\nu$ can be obtained by a convex optimization solver. After obtaining the optimal values, $(\lambda^*, \nu^*) = \operatorname{argmax}_{(\lambda,\nu)} g(\lambda, \nu)$, the policy update direction $x^*$ are calculated by $\frac{-1}{\nu^*} H^{-1}(B^T \lambda^* + g)$. Then, the policy is updated by $\psi_{t+1} = \psi_t + \beta x^*$, where $\beta$ is a step size, which can be calculated by a line search method.

# B   ABLATION STUDY ON SURROGATE FUNCTIONS

We have extended the off-policy TRPO (Meng et al., 2022) to the entropy-regularized version and re-formulated it as the SAC-style version. In this section, we evaluate the original, entropy-regularized, and SAC-style versions in the continuous control tasks of the MuJoCo simulators (Todorov et al., 2012). We use neural networks with two hidden layers with (512, 512) nodes and ReLU for the activation function. The output of a value network is linear, but the input is different; the original and entropy-regularized versions use states, and the SAC-style version uses state-action pairs. The input of a policy network is the state, the output is mean $\mu$ and std $\sigma$, and actions are squashed into $\tanh(\mu + \epsilon\sigma)$, $\epsilon \sim \mathcal{N}(0,1)$ as in SAC (Haarnoja et al., 2018). The entropy coefficient $\beta$ in the entropy-regularized and SAC-style versions are adaptively adjusted to keep the entropy above a threshold (set as $-d$ given $A \subseteq \mathbb{R}^d$). The hyperparameters for all versions are summarized in Table 1.

| Parameter | |
|---|---|
| Discount factor $\gamma$ | 0.99 |
| Trust region size $\epsilon$ | 0.001 |
| Length of replay buffer | $10^5$ |
| Critic learning rate | 0.0003 |
| Trace-decay $\lambda$ | 0.97 |
| Initial entropy coefficient $\beta$ | 1.0 |
| $\beta$ learning rate | 0.01 |

Table 1: Hyperparameters for all versions.

The training curves are presented in Figure 6. All methods are trained with five different random seeds. Since there is no importance ratio and the Q-functions directly provide policy update direction, the SAC-style version outperforms the others.

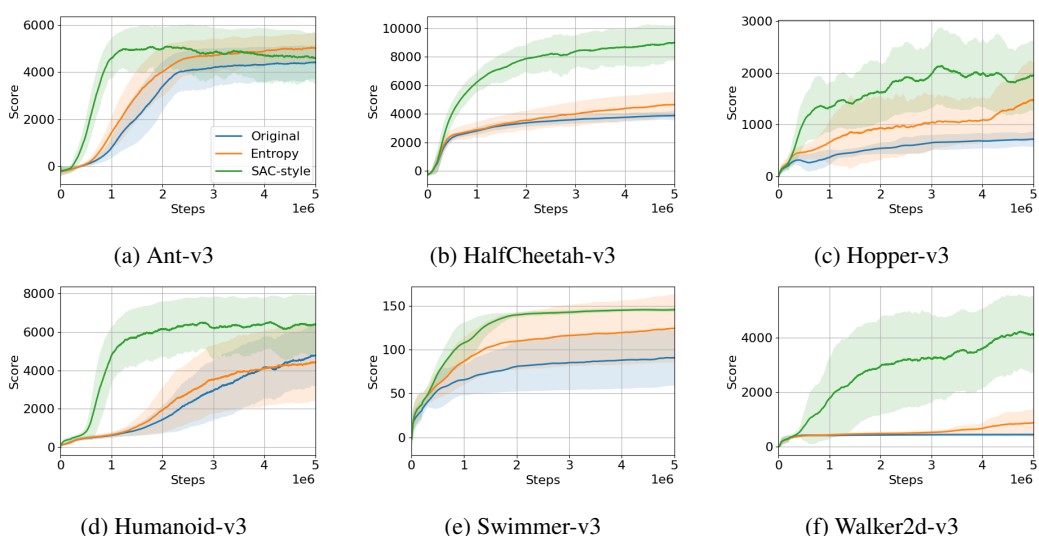

(a) Ant-v3          (b) HalfCheetah-v3          (c) Hopper-v3

(d) Humanoid-v3          (e) Swimmer-v3          (f) Walker2d-v3

Figure 6: MuJoCo training curves.

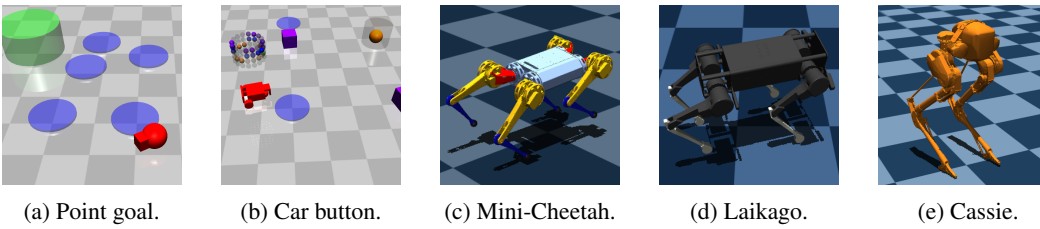

(a) Point goal.      (b) Car button.      (c) Mini-Cheetah.      (d) Laikago.      (e) Cassie.

Figure 7: (a) and (b) are Safety Gym tasks. (c), (d), and (e) are locomotion tasks.

## C  EXPERIMENTAL SETTINGS

**Safety Gym.** We use the goal and button tasks with the point and car robots in the Safety Gym environment (Ray et al., 2019), as shown in Figure 7a and 7b. The environmental setting for the goal task is the same as in Kim & Oh (2022b). Eight hazard regions and one goal are randomly spawned at the beginning of each episode, and a robot gets a reward and cost as follows:

$$R(s, a, s') = -\Delta d_{\text{goal}} + \mathbf{1}_{d_{\text{goal}} \leq 0.3},$$
$$C(s, a, s') = \text{Sigmoid}(10 \cdot (0.2 - d_{\text{hazard}})), \tag{24}$$

where $d_{\text{goal}}$ is the distance to the goal, and $d_{\text{hazard}}$ is the minimum distance to hazard regions. If $d_{\text{goal}}$ is less than or equal to $0.3$, a goal is respawned, and the number of constraint violations (CVs) is counted when $d_{\text{hazard}}$ is less than $0.2$. The state consists of relative goal position, goal distance, linear and angular velocities, acceleration, and LiDAR values. The action space is two-dimensional, which consists of $xy$-directional forces for the point and wheel velocities for the car robot.

The environmental settings for the button task are the same as in Liu et al. (2022). There are five hazard regions, four dynamic obstacles, and four buttons, and all components are fixed throughout the training. The initial position of a robot and an activated button are randomly placed at the beginning of each episode. The reward function is the same as in (24), but the cost is different since there is no dense signal for contacts. We define the cost function for the button task as an indicator function that outputs one if the robot makes contact with an obstacle or an inactive button or enters a hazardous region. We add LiDAR values of buttons and obstacles to the state of the goal task, and actions are the same as the goal task. The length of the episode is 1000 steps without early termination.

**Locomotion Tasks.** We use three different legged robots, Mini-Cheetah, Laikago, and Cassie, for the locomotion tasks, as shown in Figure 7c, 7d, and 7e. The tasks aim to control robots to follow a velocity command on flat terrain. A velocity command is given by $(v_x^{\text{cmd}}, v_y^{\text{cmd}}, \omega_z^{\text{cmd}})$, where $v_x^{\text{cmd}} \sim \mathcal{U}(-1.0, 1.0)$ for Cassie and $\mathcal{U}(-1.0, 2.0)$ otherwise, $v_y^{\text{cmd}} = 0$, and $\omega_z^{\text{cmd}} \sim \mathcal{U}(-0.5, 0.5)$. To lower the task complexity, we set the $y$-directional linear velocity to zero but can scale to any non-zero value. As in other locomotion studies (Lee et al., 2020; Miki et al., 2022), *central phases* are introduced to produce periodic motion, which are defined as $\phi_i(t) = \phi_{i,0} + f \cdot t$ for $\forall i \in \{1, ..., n_{\text{legs}}\}$, where $f$ is a frequency coefficient and is set to 10, and $\phi_{i,0}$ is an initial phase. Actuators of robots are controlled by PD control towards target positions given by actions. The state consists of velocity command, orientation of the robot frame, linear and angular velocities of the robot, positions and speeds of the actuators, central phases, history of positions and speeds of the actuators (past two steps), and history of actions (past two steps). A foot contact timing $\xi$ can be defined as follows:

$$\xi_i(s) = -1 + 2 \cdot \mathbf{1}_{\sin(\phi_i) \leq 0} \quad \forall i \in \{1, ..., n_{\text{legs}}\}, \tag{25}$$

where a value of -1 means that the $i$th foot is on the ground; otherwise, the foot is in the air. For the quadrupedal robots, Mini-Cheetah and Laikago, we use the initial phases as $\phi_0 = \{0, \pi, \pi, 0\}$, which generates trot gaits. For the bipedal robot, Cassie, the initial phases are defined as $\phi_0 = \{0, \pi\}$, which generates walk gaits. Then, the reward and cost functions are defined as follows:

$$R(s, a, s') = -0.1 \cdot (||v_{x,y}^{\text{base}} - v_{x,y}^{\text{cmd}}||_2^2 + ||\omega_z^{\text{base}} - \omega_z^{\text{cmd}}||_2^2 + 10^{-3} \cdot R_{\text{power}}),$$

$$C_1(s, a, s') = \mathbf{1}_{\text{angle} \geq a}, \ C_2(s, a, s') = \mathbf{1}_{\text{height} \leq b}, \ C_3(s, a, s') = \sum_{i=1}^{n_{\text{legs}}} (1 - \xi_i \cdot \hat{\xi}_i)/(2 \cdot n_{\text{legs}}), \tag{26}$$

where the power consumption $R_{\text{power}} = \sum_i |\tau_i v_i|$, the sum of the torque times the actuator speed, is added to the reward as a regularization term, $v_{x,y}^{\text{base}}$ is the $xy$-directional linear velocity of the base frame of robots, $\omega_z^{\text{base}}$ is the $z$-directional angular velocity of the base frame, and $\hat{\xi} \in \{-1,1\}^{n_{\text{legs}}}$ is the current feet contact vector. For balancing, the first cost indicates whether the angle between the $z$-axis vector of the robot base and the world is greater than a threshold ($a = 15°$ for all robots). For standing, the second cost indicates the height of CoM is less than a threshold ($b = 0.3, 0.35, 0.7$ for Mini-Cheetah, Laikago, and Cassie, respectively), and the last cost is to check that the current feet contact vector $\hat{\xi}$ matches the pre-defined timing $\xi$. The length of the episode is 500 steps. There is no early termination, but if a robot falls to the ground, the state is frozen until the end of the episode.

**Hyperparameter Settings.** The structure of neural networks consists of two hidden layers with $(512, 512)$ nodes and ReLU activation for all baselines and the proposed method. The input of value networks is state-action pairs, and the output is the positions of atoms. The input of policy networks is the state, the output is mean $\mu$ and std $\sigma$, and actions are squashed into $\tanh(\mu + \epsilon\sigma)$, $\epsilon \sim \mathcal{N}(0,1)$. We use a fixed entropy coefficient $\beta$. The trust region size $\epsilon$ is set to $0.001$ for all trust region-based methods. The overall hyperparameters for the proposed method can be summarized in Table 2. Since the range of the cost is $[0, 1]$, the maximum discounted cost sum is $1/(1 - \gamma)$. Thus, the limit

| Parameter | Safety Gym | Locomotion |
|---|---|---|
| Discount factor $\gamma$ | 0.99 | 0.99 |
| Trust region size $\epsilon$ | 0.001 | 0.001 |
| Length of replay buffer | $10^5$ | $10^5$ |
| Critic learning rate | 0.0003 | 0.0003 |
| Trace-decay $\lambda$ | 0.97 | 0.97 |
| Entropy coefficient $\beta$ | 0.0 | 0.001 |
| The number of critic atoms $M$ | 25 | 25 |
| The number of target atoms $M'$ | 50 | 50 |
| Constraint conservativeness $\alpha$ | 0.25, 0.5, and 1.0 | 1.0 |
| Limit value $d_k$ | $0.025/(1-\gamma)$ | $[0.025, 0.025, 0.4]/(1-\gamma)$ |
| Slack coefficient $\zeta$ | - | $\min_k d_k = 0.025/(1-\gamma)$ |
| Reward weights $w_i$ | - | $[5.19, 3.71, 0.34]$ |

Table 2: Hyperparameter settings for the Safety Gym and locomotion tasks.

value is set by target cost rate times $1/(1 - \gamma)$. For the locomotion tasks, the third cost in (26) is designed for foot stamping, which is not essential to safety. Hence, we set the limit value to near the maximum (if a robot does not stamp, the cost rate becomes $0.5$). The reward weights are also presented in Table 2, which are optimized using the existing Bayesian optimization tool. In addition, baseline methods use multiple critic networks for the cost function, such as target (Yang et al., 2021) or square value networks (Kim & Oh, 2022a). To match the number of network parameters, we use two critics as an ensemble, as in Kuznetsov et al. (2020).

**Tips for Hyperparameter Tuning.**

- Discount factor $\gamma$, Critic learning rate: Since these are commonly used hyperparameters, we do not discuss these.

- Trace-decay $\lambda$, Trust region size $\epsilon$: The ablation studies on these hyperparameters are presented in Appendix D.4. From the results, we recommend setting the trace-decay to $0.95 \sim 0.99$ as in other TD($\lambda$)-based methods (Precup et al., 2000). Also, the results show that the performance is not sensitive to the trust region size. However, if the trust region size is too large, the approximation error increases, so it is better to set it below $0.003$.

- Entropy coefficient $\beta$: This value is fixed in our experiments, but it can be adjusted automatically as done in SAC (Haarnoja et al., 2018).

- The number of atoms $M, M'$: Although experiments on the number of atoms did not performed, performance is expected to increase as the number of atoms increases, as in other distributional RL methods Dabney et al. (2018a).

- Length of replay buffer: The effect of the length of the replay buffer can be confirmed through the experimental results from an off policy-based safe RL method (Kim & Oh,

2022a). According to that, the length does not impact performance unless it is too short. We recommend setting it to 10 to 100 times the collected trajectory length.

- Constraint conservativeness $\alpha$, limit value $d_k$: If the cost sum follows a Gaussian distribution, the mean-std constraint becomes the CVaR constraint. Then, the probability of the worst case can be controlled by adjusting $\alpha$. For example, if we set $\alpha = 0.125$ and $d = 0.03/(1 - \gamma)$, the mean-std constraint enforces the probability that the average cost is less than 0.03 during an episode greater than $95\% = \Phi(\phi(\Phi^{-1}(\alpha))/\alpha)$. Through this meaning, proper $\alpha$ and $d_k$ can be found.

- Slack coefficient $\zeta$: As mentioned at the end of Section 3.3, it is recommended to set this coefficient as large as possible. Since $d_k - \zeta$ should be positive, we recommend setting $\zeta$ to $\min_k d_k$.

- Reward weights: These are used when defining the reward function for traditional RL methods, so these are not hyperparameters of our method.

In conclusion, most hyperparameters are not sensitive, so few need to be optimized. It seems that $\alpha$ and $d_k$ need to be set based on the meaning described above. Additionally, if the approximation error of critics is significant, the trust region size should be set smaller.

# D EXPERIMENTAL RESULTS

## D.1 SAFETY GYM

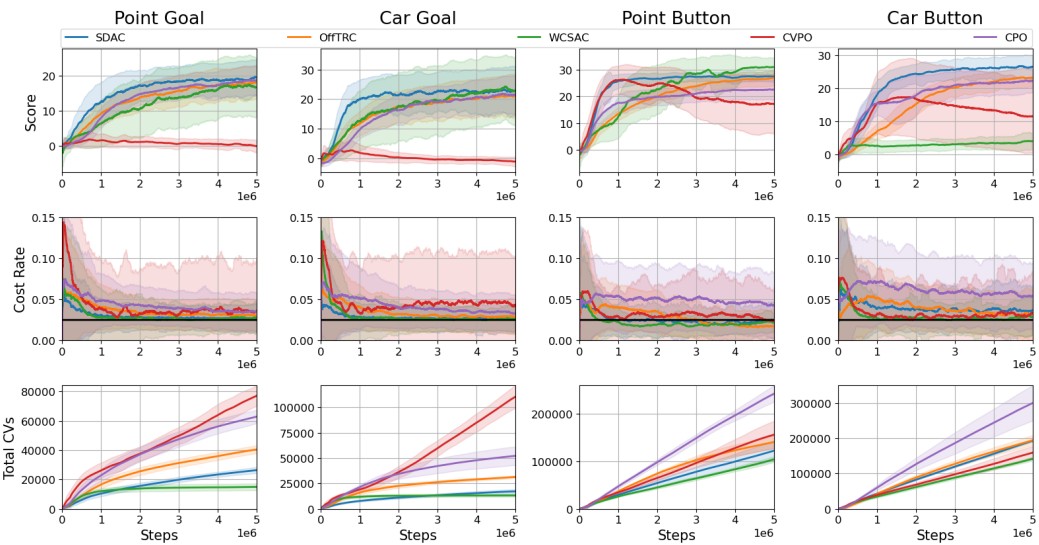

Figure 8: Training curves of mean constrained algorithms for the Safety Gym tasks. The solid line and shaded area represent the average and std values, respectively. The solid black lines in the second row indicate limit values. All methods are trained with five random seeds.

In this section, we present the training curves of the Safety Gym tasks separately according to the conservativeness of constraints for better readability. Figure 8 shows the training results of the mean constrained and mean-std constrained algorithms with $\alpha = 1.0$. Figures 9 and 10 show the training results of the mean-std constrained algorithms with $\alpha = 0.25$ and $0.5$, respectively. In Figure 8, it can be observed that the score of SDAC has the fastest convergence speed and that the cost rates also converge to the limit values quickly. Observing that all the other methods show the highest total CVs in the car button task, this task is challenging to meet the constraint. Thus, SDAC also has a higher cost rate for the car button than other tasks. In addition, since decreasing $\alpha$ makes the constraints more conservative, the cost rates and the number of total CVs of SDAC are reduced in Figures 9 and 10. For $\alpha = 0.25$ and $0.5$, SDAC shows the highest scores and the lowest number of CVs in all tasks.

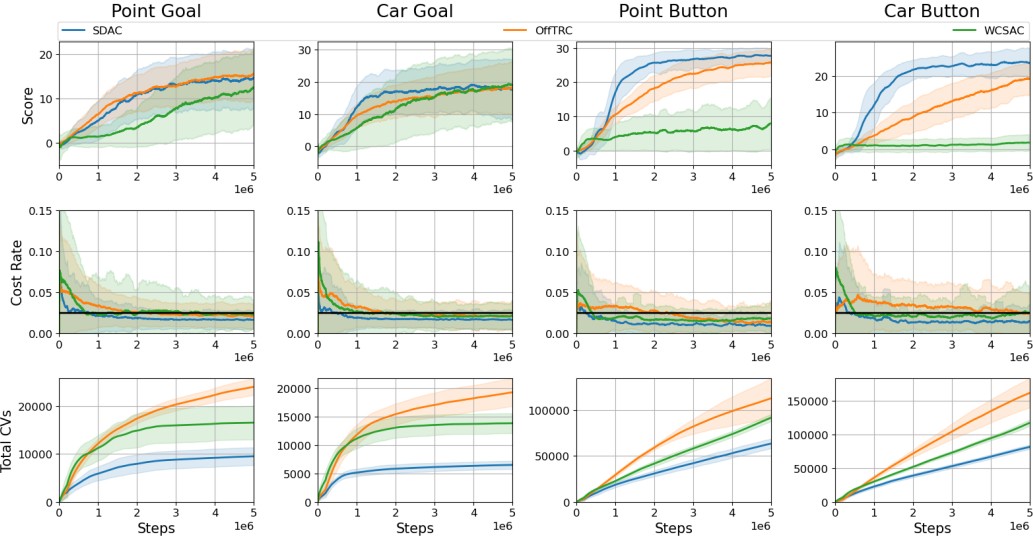

Figure 9: Training curves of mean-std constrained algorithms with $\alpha = 0.5$ for the Safety Gym.

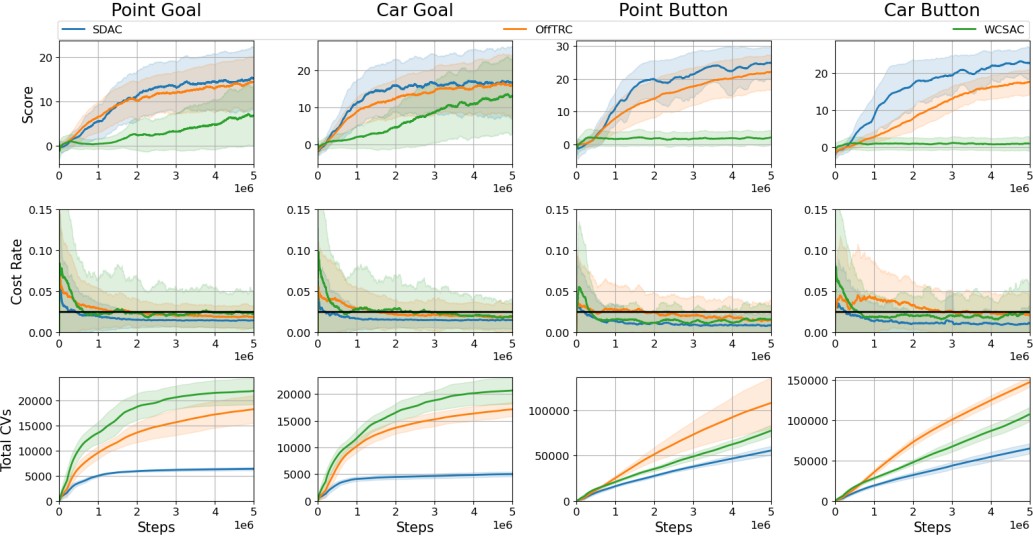

Figure 10: Training curves of mean-std constrained algorithms with $\alpha = 0.25$ for the Safety Gym.

## D.2 LOCOMOTION TASKS

We present the training curve for the locomotion tasks in Figure 11, where the true reward is defined in (26). The traditional RL baselines are trained with the reward $\bar{R} = (R - \sum_{i=1}^{3} w_i C_i)/(1 + \sum_{i=1}^{3} w_i)$, and the weights are described in Section C. The training curve contains graphs of the three cost rates and the true reward sum.

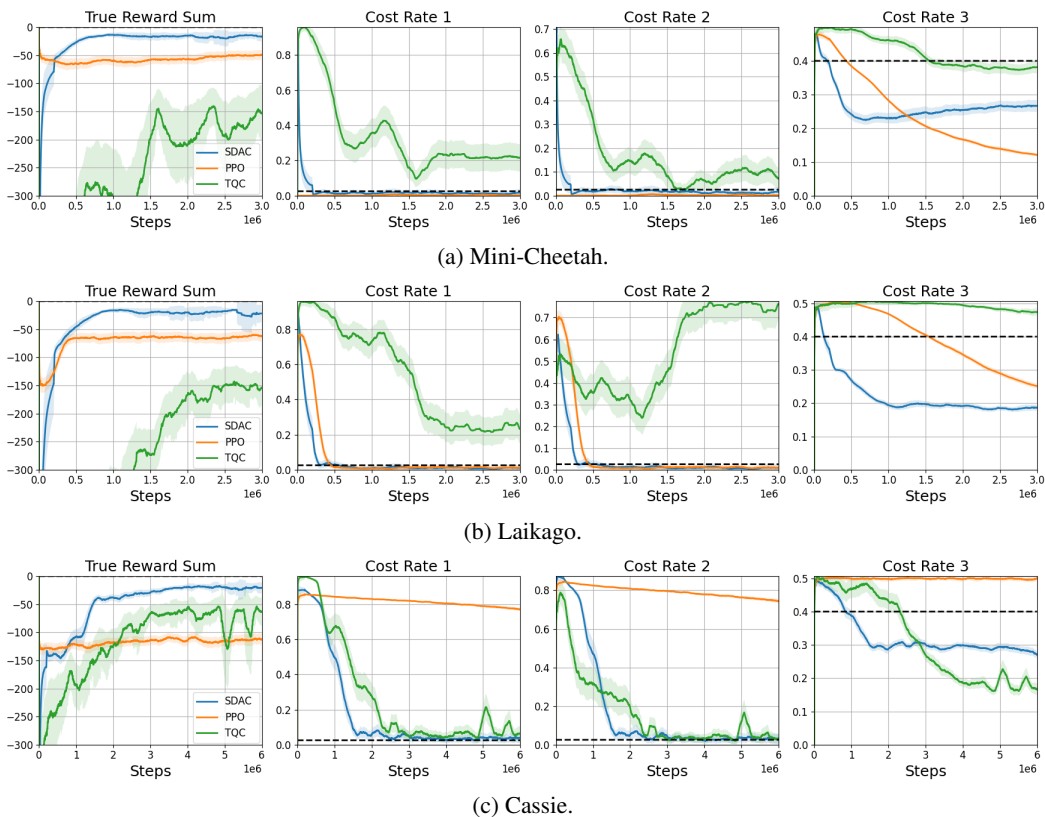

Figure 11: Training curves of the locomotion tasks. The black dashed lines show the limit values used for the safe RL methods. The solid line represents the average value, and the shaded area shows one-fifth of the std value. All methods are trained with five different random seeds.

### D.3 ABLATION STUDY ON COMPONENTS OF SDAC

We experiment with variations of SDAC to examine the effectiveness of each component. SDAC has two main components, SAC-style surrogate functions and distributional critics. We call SDAC with only distribution critics, *SDAC-Dist*, and SDAC with only SAC-style surrogates, *SDAC-Q*. If all components are absent, SDAC is identical to OffTRC (Kim & Oh, 2022a). The variants are trained with the point goal task of the Safety Gym, and the training results are shown in Figure 12. SDAC-Q lowers the cost rate quickly but shows the lowest score. SDAC-Dist shows scores similar to SDAC, but the cost rate converges above the limit value $0.025$. In conclusion, SDAC can efficiently satisfy the safety constraints through the SAC-style surrogates and improve score performance through the distributional critics.

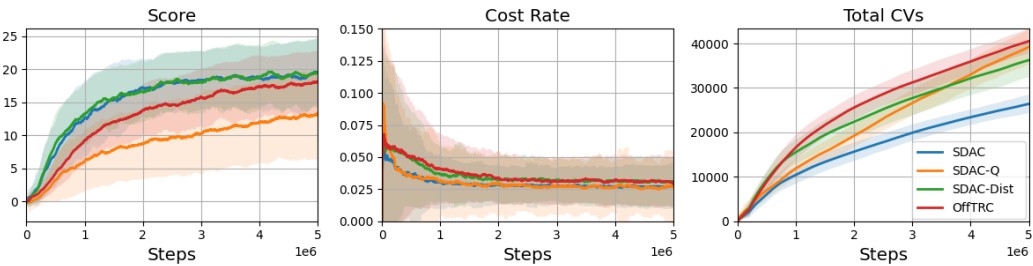

Figure 12: Training curves of variants of SDAC for the point goal task.

### D.4 ABLATION STUDY ON HYPERPARAMETERS

To check the effects of the hyperparameters, we conduct ablation studies on the trust region size and entropy coefficient. The results are shown in Figure 13. From the entropy coefficient results, it can be seen that excessive exploration causes the constraint to be violated. Thus, the entropy coefficient should be adjusted cautiously, or it can be better to set the coefficient to zero. Since Theorem 1 shows that the estimation error of the surrogates is proportional to the trust region size, it can be observed from Figure 13b that the number of CVs increases with the size of the trust region due to the estimation error.

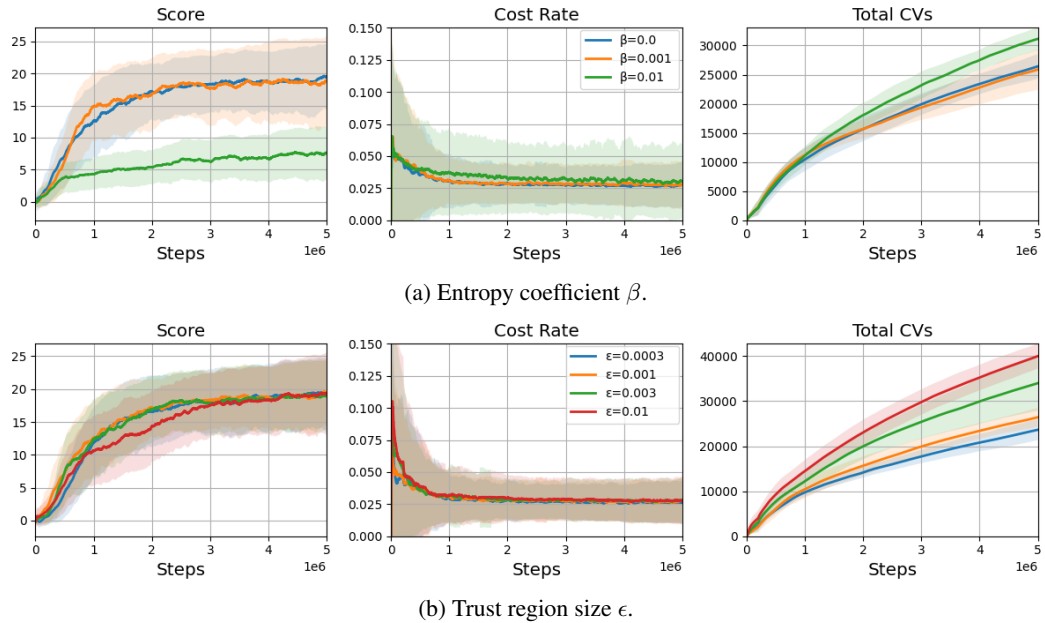

Figure 13: Training curves of SDAC with different hyperparameters for the point goal task.

## E COMPUTATIONAL COST ANALYSIS

In this section, we analyze the computational cost of the gradient integration method. The proposed gradient integration method has three subparts. First, it is required to calculate policy gradients of each cost surrogate, $g_k$, and $H^{-1}g_k$ for $\forall k \in \{1, 2, ..., K\}$, where $H$ is the Hessian matrix of the KL divergence. $H^{-1}g_k$ can be computed using the conjugate gradient method, which requires only a constant number of back-propagation on the cost surrogate, so the computational cost can be expressed as $K \cdot O(\text{BackProp})$.

Second, the quadratic problem in Section 3.3 is transformed to a dual problem, where the transformation process requires inner products between $g_k$ and $H^{-1}g_m$ for $\forall k, m \in \{1, 2, ..., K\}$. The computational cost can be expressed as $K^2 \cdot O(\text{InnerProd})$.

Finally, the transformed quadratic problem is solved in the dual space $\in \mathbb{R}^K$ using a quadratic programming solver. Since $K$ is usually much smaller than the number of policy parameters, the computational cost almost negligible compared to the others. Then, the cost of the gradient integration is $K \cdot O(\text{BackProp}) + K^2 \cdot O(\text{InnerProd}) + C$. Since the back-propagation and the inner products is proportional to the number of policy parameters $|\psi|$, the computational cost can be simplified as $O(K^2 \cdot |\psi|)$.

