# OpenReview forum: "SDAC: Efficient Safe Reinforcement Learning with Low-Biased Distributional Actor-Critic"
_ICLR.cc/2023/Conference — Submitted to ICLR 2023_

### Official Review · Reviewer_FqLh · 2022-10-21

**Confidence:** 3
**Correctness:** 3
**Technical Novelty And Significance:** 3
**Empirical Novelty And Significance:** 3
**Recommendation:** 6

**Clarity, Quality, Novelty And Reproducibility:**

In general, the paper is clear. However, it is advisable to have a look at the papers the proposed SDAC method is built on top of.
The technical quality is high, and the authors seem to have the ambition to use sota-methods for any part of their method.
The paper does provide code and provides hyperparameters and settings in the appendix. Thus, reproducing the results should be no major problem,
The novelty of the approach is kind of hard to assess. The authors have engineered a well-performing safe RL method. However, the contributions do not seem to follow a single general idea but are tailor-made to make the combination to trust-region-based safe RL and distributional methods work.

**Strength And Weaknesses:**

strong points:
* The paper combines and extends several SOTA techniques.
* Results indicate a superior trade-off between constraint violations and performance

weak points:
* The paper's contributions are not easily distinguishable from the combined basis methods.
* it is unclear how the contributions relate to each other. Some of them seem disconnected and not really specific to safe RL. In particular, using a distributional critic seems valid for any AC method.
* Some of the methods are not well-motivated. For example, I did not get why replacing the advantage function with the q-function is beneficial.

After reading the author's comments and the revised paper, explaining why these methods address open problems of constrained RL with trust regions is more precise.


**Summary Of The Paper:**

The paper proposes a method for safe RL or more general constrained RL. Constraints are modelled by the mean-std method. As RL algorithm the authors use TRPO and use a multi-step distributional critic.
Apart from combining these methods, the paper seems to have three independent contributions:
1. The authors extend the distributional RL approach based on quantile regression and Wasserstein distance to the multistep setting.
2. Then replace the advantage function with the q-function in estimating the trust region.
3. Finally, they propose a gradient-based method to find starting policies in the trust region.
Experiments are conducted on the safety Gym environment and compare to other safe RL approaches w.r.t. constraint violations and performance.

**Summary Of The Review:**

The paper proposes a well-performing safe RL-method which combines several recent approaches such as trust-region-based safe RL and distributional critics. Experimental results are also quite convincing.
On the other hand, some of the contributions are not well-motivated. For example, the authors claim that it is a good idea to use q-values instead of the advantage function. However,  as the advantage function is the q-function minus the value function and it was introduced to lower the variance, it is kind of unclear why this is done. Later, they add entropy regularization which makes more sense, but it is not very exciting as it is a well-known technique.
In general, many improvements seem to be a relatively straightforward combination of existing techniques and a better explanation on why the proposed extension yield a major technical contribution would be desireable.

---

> ### Author Response · Authors · 2022-11-13
> **Response to Reviewer FqLh (Part 2)**
>
> ### **Q3. Reproducibility.**
>
> We have attached the source code in the supplementary material, so please refer to the code.
>
> ### **References**
>
> [1] Joshua Achiam. Spinning Up in Deep Reinforcement Learning. 2018.
>
> [2] Antonin Raffin, Ashley Hill, Adam Gleave, Anssi Kanervisto, Maximilian Ernestus, and Noah
> Dormann. Stable-baselines3: Reliable reinforcement learning implementations. Journal of
> Machine Learning Research, 22(268):1–8, 2021.

---

> > ### Comment · Reviewer_FqLh · 2022-11-16
> > **section on reproducibility revised**
> >
> > Thanks for pointing this out. I have revised my review, attesting you the good reproducibility of your results now.

---

> ### Author Response · Authors · 2022-11-13
> **Response to Reviewer FqLh (Part 1)**
>
> We thank reviewer FqLh for the thorough review that provided an opportunity to improve our manuscript.
> We respond to reviewer FqLh's suggestions and comments below.
>
> ### **Q1. The contributions are not easily distinguishable and not well-motivated.**
> > "The paper's contributions are not easily distinguishable from the combined basis methods. it is unclear how the contributions relate to each. Some of them seem disconnected and not really specific to safe RL. In particular, using a distributional critic seems valid for any AC method. Some of the methods are not well-motivated. For example, I did not get why replacing the advantage function with the q-function is beneficial."
>
> Thanks for the comment, and we agree that the paper could have delivered the motivation of each contribution better.
> We have revised the structure and contents of the introduction section to clarify what contributions are proposed and why each contribution is necessary.
> Please see the revised version.
>
>
> - **Subquestion 1**
>     > "In particular, using a distributional critic seems valid for any AC method."
>
>     We agree with the reviewer's comment, but we would like to say that our contribution is not using a distributional critic but training a distributional critic with a low estimation bias using the proposed target distribution.
>
>     In addition, we can say that the proposed target distribution is a highly related contribution to safe RL.
>     Satisfying constraints is the top priority of safe RL.
>     To this end, it is required to check that the current policy satisfies constraints, where the value of the constraints can be estimated by critics.
>     Then if the estimation biases of the critics are low enough, we can update the policy in the correct direction.
>     However, there is a lack of techniques to lower the bias of distributional critics.
>     Thus, we propose the TD($\lambda$) target distribution, which can control the bias-variance tradeoff, and empirically show that it can lower the biases, as observed in Section 5.3.
>
> - **Subquestion 2**
>     > "I did not get why replacing the advantage function with the q-function is beneficial."
>
>     Thanks for the comment, and we have added an explanation of why Q-functions are beneficial to the advantage functions in Section 3.2.
>     Like the reviewer's comment, the advantage function is normalized by subtracting value function from Q-function, so it gives an effect to decrease the variance of the policy gradient.
>     However, in the off-policy setting, the deviation of the importance ratio becomes significant, so the merit of the advantage functions may be ignored (please refer to the surrogate function, $J^{\mu,\pi}(\pi')=\mathbb{E}_{a\sim\mu}[\frac{\pi'(a|s)}{\mu(a|s)}A^{\pi}(s,a)]$).
>     Second, the advantage function only gives scalar information about whether a sampled action is proper.
>     In contrast, the Q-function directly gives the direction in which the action should be updated, so more information can be obtained using Q-function.
>     Also, we have empirically compared the two versions of the surrogate on the MuJoCo tasks, and the experimental results are presented in Appendix B.
>     According to the results, the SAC-style surrogate outperforms the advantage-based surrogate.
>
> In addition, the gradient integration method is also a noteworthy contribution.
> In safe RL, policies can be updated in the direction of satisfying constraints using a safe policy update rule, and there are two general safe policy update rules: the trust region-based method and the Lagrangian method.
> However, the Lagrangian method cannot theoretically guarantee satisfying constraints during training, and the trust region-based method lacks a handling method for infeasible starting cases under multiple constraint settings (but there is a naive approach, please refer to Section 3.3).
> For more details on the infeasible starting cases, it means that no policy satisfies constraints within the trust region of the given policy due to initial policy settings.
> If the initial policy does not satisfy constraints, it should be projected on the feasible policy space.
> To this end, we propose the gradient integration method, which handles the infeasible starting cases for multiple constraint settings, to use the trust region-based method.
> We can summarize it as follows:
> || Can guarantee that constraints are satisfied during training? | Can handle multiple constraints? |
> |-|-|-|
> |Lagrangian|❌|✅|
> |Trust region|✅|⚠️|
> |**SDAC** (proposed)|✅|✅|
>
> ### **Q2. The entropy regularization is a well-known technique.**
>
> We agree with the reviewer's comment.
> Many RL algorithms add the entropy regularization term to policy loss to encourage effective exploration as a heuristic technique.
> In contrast, we show that the proposed entropy-regularized surrogate function has a bound within a trust region in Theorem 1, so the proposed surrogate can still be applied to the trust region method, which can be considered a contribution.

---

> > ### Comment · Reviewer_FqLh · 2022-11-16
> > **thank you for your clarification.**
> >
> > Dear authors,
> >
> > thank you for providing this information. I now understand better why using TD(\lambda) is particularly useful for the safe RL setting.
> > I will reread the paper to understand the other explanations fully.

---

### Official Review · Reviewer_97Q2 · 2022-10-25

**Confidence:** 4
**Clarity, Quality, Novelty And Reproducibility:** Clarity, Quality, Novelty, and Reprod…
**Correctness:** 2
**Technical Novelty And Significance:** 2
**Empirical Novelty And Significance:** Not applicable
**Recommendation:** 3

**Strength And Weaknesses:**

Strength
1. introduce a TD(λ) target distribution combining multiple-step distributions to estimate constraints with low biases
2. present a memory-efficient method to approximate the TD(λ) target distribution using quantile distributions
3. propose novel surrogate functions which use the reparameterization trick with Q-functions to increase the efficiency of the trust region method,

Weaknesses

1. I don't understand what figure 1 is trying to say, could the author be more clear?
2. The authors believe that Figure 3 can be clearly represented, "the result can be interpreted as excellent", but from this figure, I have no way to see the cost limit, as well as fine metrics, intelligent roughly, the method of this paper is roughly put in the upper left, but Point Goal and Car Goal seem to overlap in the upper left.
3. I carefully read through the experimental results of the article, and in Appendix D.1, Figure 8, in Point Goal and Point Button, WCSAC is significantly better than the SDAC proposed in this paper in terms of score, cost rate, and Total CVs, and this seems to be consistent with the Safe RL perspective. From the Safe RL perspective, this seems to be in great conflict with what is discussed in the abstract, "From extensive experiments, the proposed method shows minimal constraint violations while achieving high returns compared to existing safe RL methods."
    I suggest that the authors should rework Figure 3 or put a more refined result, Figure 8, in the text, because the ambiguous and unrefined Figure 3 leads to a misunderstanding of the performance of the algorithm.
4. In Figure 8, why is there if a big difference between cvpo in Button's task and Goal's task? cvpo's reward in Goal's task is even negative, while it does have a considerable reward in Button's task.
5. As far as I know, \alpha is the discount factor corresponding to cost, so why the comparison in Figure 9 and Figure 10 lack CPO and CVPO.
Based on the above doubts, I have doubts about the contradictory claims in this article and the results of the experiments, and the authors don't seem to disclose their code in the appendix material. Also based on the list of hyperparameters provided by the authors (and no other baseline details of the experimental parameters are provided), I think the reproducibility of the experimental results of this work is very low.


**Summary Of The Paper:**

The author develop an efficient trust region-based safe RL algorithm with multiple constraints.
1. introduce a TD(λ) target distribution combining multiple-step distributions to estimate constraints with low biases
2. present a memory-efficient method to approximate the TD(λ) target distribution using quantile distributions
2. to increase the efficiency of the trust region method, we propose novel surrogate functions which use the reparameterization trick with Q-functions

**Summary Of The Review:**

I have many doubts about the experimental results and opinions in this article, and I hope the authors can explain them in detail.

---

> ### Author Response · Authors · 2022-11-13
> **Response to Reviewer 97Q2 (Part 2)**
>
> - **[Answer to question 3]**  The results of the point goal task in Figure 8 shows that the cost rates of WCSAC and SDAC are close to the limit value, which can be interpreted as both are satisfied the constraint.
> However, WCSAC shows lower total CVs and scores compared to SDAC, which means that SDAC is better in terms of the score, and WCSAC is better in terms of safety, but both satisfy the constraint.
> For this reason, it is difficult to conclude that WCSAC is better than SDAC.
> There is one exception.
> For the point button task, WCSAC shows higher scores and lower total CVs than SDAC, so WCSAC is better than SDAC for that task.
> Additionally, it is essential to check training trends according to $\alpha$ since WCSAC and SDAC are mean-std constrained safe RL methods.
> In Figures 9 and 10, which show the experimental results for different $\alpha$, SDAC shows the highest score and the lowest total CVs for all tasks.
> In conclusion, it can be said that SDAC shows better performance than WCSAC in overall experiments, although WCSAC shows better results for the point button task with $\alpha=1.0$.
> However, as the reviewer mentioned, we judged that the sentence in the abstract conflicts with some results, so we have refined it to "The proposed method with risk-averse constraints shows minimal constraint violations while achieving high returns."
> - **[Answer to question 5]** We think there seems to be some misunderstanding about the meaning of $\alpha$, and we hope the above explanation has delivered the meaning well.
>
> ### **Q3. In Figure 8, why is there if a big difference between cvpo in Button's task and Goal's task?**
>
> We can say that the goal tasks are more challenging to solve than the button tasks since the scores of button tasks are larger than the goal tasks in the results of the other safe RL methods.
> Actually, we lowered the difficulty of the button tasks by fixing the position and path of the obstacles, whereas obstacles are randomly spawned in the goal tasks.
> Therefore, CVPO performs worse for the goal tasks than button tasks.
> Also, the score of CVPO tends to rise and fall for both button and goal tasks.
> We infer that the cause is that information loss occurs during the EM steps in the policy update of CVPO.
> CVPO finds a nonparametric policy that satisfies constraints in E-step and fits the parametric policy to the nonparametric policy in M-step.
> However, information about the constraints can be lost during the M-step.
>
> ### **Q4. Reproducibility.**
>
> We have attached the source code in the supplementary material, so please refer to the code.
>
> ### **References**
>
> [1] The investopia team. (2022, September 20). *Pareto Efficiency Examples and Production Possibility Frontier*. https://www.investopedia.com/terms/p/pareto-efficiency.asp

---

> > ### Comment · Reviewer_97Q2 · 2022-11-18
> > **Response to authors**
> >
> > From my point of view, I sincerely suggest that the author modify the rebuttal version by adding content highlighting or changing the color representation, so that it is clear what content has been modified, instead of making me compare it with the original version again.
> > Thank you for the clarifications and updates. I will carefully read the new version of the paper.

---

> > > ### Author Response · Authors · 2022-11-18
> > > **Highlighted version**
> > >
> > > Thanks for the recommendation. As the reviewer said, we re-uploaded the highlighted manuscript.

---

> > > ### Author Response · Authors · 2022-11-30
> > > **Further discussion**
> > >
> > > Dear Reviewer 97Q2,
> > >
> > > Thanks again for your thoughtful review.
> > > We would like to ask if our answers to the experimental results are sufficient and whether the revised version has sufficiently reflected the reviewer's comment.
> > > Any further questions or discussions are welcome.
> > >
> > > Sincerely,
> > >
> > > Paper authors.

---

> > > ### Author Response · Authors · 2022-12-12
> > > **Last day reminder**
> > >
> > > Dear Reviewer 97Q2,
> > >
> > > Thanks again for reviewing our paper.
> > > Today is the last day of the review process, and we look forward to your response to the revised paper.
> > >
> > > Thank you,
> > >
> > > Paper authors.

---

> ### Author Response · Authors · 2022-11-13
> **Response to Reviewer 97Q2 (Part 1)**
>
> We thank reviewer 97Q2 for the thorough review on our work.
> We respond to reviewer 97Q2's suggestions and comments below.
>
> ### **Q1. Make Figure 1 clearer.**
>
> Thanks for the comments; we modified Figure 1 and added more explanations in the caption.
> We visualize the calculation process of the proposed target distribution, which is used to train distributional critics, in Figure 1.
> In particular, Figure 1 shows the calculation process at time step $t$ , and the target distribution can be recursively calculated in the order of $T, T-1, ..., 1$, where $T$ is the episode length.
> Please refer to the modified manuscript.
>
> ### **Q2. Experimental results from Figure 3, 9, 8, and 10.**
> >"2. The authors believe that Figure 3 can be clearly represented, "the result can be interpreted as excellent", but from this figure, I have no way to see the cost limit, as well as fine metrics, intelligent roughly, the method of this paper is roughly put in the upper left, but Point Goal and Car Goal seem to overlap in the upper left."
>
> > "3. I carefully read through the experimental results of the article, and in Appendix D.1, Figure 8, in Point Goal and Point Button, WCSAC is significantly better than the SDAC proposed in this paper in terms of score, cost rate, and Total CVs, and this seems to be consistent with the Safe RL perspective."
>
> > "5. As far as I know, \alpha is the discount factor corresponding to cost, so why the comparison in Figure 9 and Figure 10 lack CPO and CVPO."
>
> Thanks for the comments, but we think there is a misunderstood part, so we will clarify the explanation of the figures first.
> SDAC (the proposed method), WCSAC, and OffTRC use the mean-std constraint, whose conservativeness can be adjusted by a hyperparameter $\alpha\in(0,1]$ (If $\alpha$ is close to zero, the weight of std in the mean-std constraint increases, so the constraint becomes conservative. If $\alpha$ is set to one, the weight of std is zero, so the constraint is the same as the mean constraint).
> The other safe RL baseline methods use the mean constraint.
> To show the training trends according to $\alpha$, we have conducted experiments with several $\alpha$ values and presented the experimental results in Figure 8, 9, and 10.
> Figure 8 shows the result of all safe RL methods with the mean constraint, which means that $\alpha$ for OffTRC, WCSAC, and SDAC is set to one.
> Figure 9 and 10 show the results of mean-std constrained safe RL methods with $\alpha=0.5$ and $0.25$, respectively.
> To show the combined results of the three figures, we have presented a graph of the reward sum versus the total number of constraint violations (CVs) instead of the training curves in Figure 3.
>
> - **[Answer to question 2]** The safe RL problem can be considered a dual objective problem since the problem is to maximize the reward sum and minimize the number of CVs.
> Therefore, the graph of reward sums versus the total number of CVs is an appropriate representation to show the dual objective performance.
> In that graph (Figure 3), if a point is located in the upper left or is not dominated by the other points, we can say the point is Pareto efficient [1], and all points of the proposed method are Pareto efficient except for the result of the point button task with $\alpha=1.0$.
> However, as the reviewer said, it is required to check that the constraints are satisfied, which means the cost sums are below the limit values. Thus, we have attached the training curves in Appendix D.1 (Figure 8, 9, and 10).

---

### Official Review · Reviewer_SPhH · 2022-10-27

**Confidence:** 3
**Correctness:** 2
**Technical Novelty And Significance:** 3
**Empirical Novelty And Significance:** 3
**Recommendation:** 5

**Clarity, Quality, Novelty And Reproducibility:**

The biggest weakness of the paper is, at least to me, the clarity of writing. While I believe that the approach is sensible (good quality), and that it is novel, I find the paper extremely hard to follow. I had to read the introduction at least three times to have an idea of the approach, and the tense technical description does not help me. I think with more structure and more intuition, the paper can definitely be improved.

I have some specific questions, and I am happy to re-evaluate my judgement after the discussion.

— computing the target distribution in an off-policy setting sounds a bit like ‘safe policy improvement’ (SPI), coined by Thomas et al. This offline RL technique estimates a better policy than the data-collecting behavior policy from a limited amount of data. Some clarification or comparison there would be nice.

— Is the overall methods ‘safe’ or reliable, that is, are we guaranteed to get a final policy that is safe? Can anything be said about a trade-off between safety and rewards?

— In the first part of the experiments, I do not understand the multiple constraint setting. Are there even multiple constraints? And if not, why is the method performing so much better than state-of-the-art baselines, who are supposed to perform well on the single constraint setting.

— Could it be an interesting experiment to shape a cost function that aims to capture the multiple constraints, and then show the differences in safety?


**Strength And Weaknesses:**

++ Important problem fitting for ICLR
++ Novel approach with multiple cost functions
++ Interesting actor-critic technique

— The paper is at parts very hard to follow
— Unclear utility of multiple cost functions in the Safety Gym environments


**Summary Of The Paper:**

This paper concerns safe (deep) reinforcement learning. In particular, the authors present an approach that take the constrained MDP setting, where on top of a reward function, a cost function is given that conveniently reflects safety concerns. The learning goal is then to maximize the reward, while the cost needs to stay below a certain thresholed. In this work, the authors employ actually a set of cost functions. The actual approach presented here entails a distributional RL method that uses trust region optimization, like in other prior works. Here, however, the multipe constraints cause problems, and the authors use (1) a target distribution to estimate constraint biases, and (2) something called the ‘parameterization trick’ to develop a dedicated actor-critic techniqe. The approaches are evaluated by means of several environments from the Safety Gym, and further robotics experiments.


**Summary Of The Review:**

Good approach, clarity can be improved.

---

> ### Author Response · Authors · 2022-11-13
> **Response to Reviewer SPhH (Part 2)**
>
> ### **Q4. Can anything be said about a trade-off between safety and rewards?**
>
> Yes, there can be a trade-off between safety and rewards.
> If we lower the limit value, $d_k$, or the constraint conservativeness, $\alpha$, to increase safety, satisfying constraints becomes challenging.
> As a result, the reward sum can be reduced, which can be interpreted as a trade-off between safety and rewards.
> In our paper, we have presented the experimental results with different $\alpha$ in Figure 8, 9, and 10.
> If $\alpha \in (0,1]$ is close to 0, the mean-std constraint becomes more conservative since the weight of std in the mean-std constraint increases.
> From the figures, it can be observed that as the value of $\alpha$ decreases, both the cost ratio and the reward sums decrease, which means that the policy becomes safer, but the performance decreases.
>
> ### **Q5. Multiple constraint setting.**
> > "In the first part of the experiments, are there even multiple constraints? And if not, why is the method performing so much better than state-of-the-art baselines, who are supposed to perform well on the single constraint setting."
>
> Thanks for the comment, and we have noticed that we still need to show how multiple constraint settings are applied.
> Thus, we have added a description of the constraint settings for each experiment in the experimental section.
>
> Here, we would like to explain more details.
> There is only one constraint in all Safety Gym tasks, which is equally applied to all safe RL methods.
> In the locomotion tasks, there are three constraints, which are well described in Appendix C.
> To explain why the proposed method performs better than other baselines, we would like to mention our contributions first.
> We propose the SAC-like surrogates to increase the efficiency of the trust region method.
> Since the SAC-like surrogates use Q-functions instead of advantage functions with importance sampling, it can avoid instability of training due to the significant variance of the importance ratio.
> As a result, our method can perform better than other baselines.
> In addition, it has been confirmed that using distributional critics improves the performance through the ablation study conducted in Appendix D.3.
>
> ### **Q6. Additional experiment.**
> > "Could it be an interesting experiment to shape a cost function that aims to capture the multiple constraints, and then show the differences in safety?"
>
> Thanks for the comment; we agree that the proposed experiment can capture interesting points.
> We want to conduct the proposed experiment, but integrating multiple cost functions into a single cost causes cumbersome work, as in reward engineering.
> If there are multiple cost functions, we can adjust the safety level of each constraint independently by adjusting the limit values, $d_k$.
> However, if the cost functions are integrated into a single cost, it is difficult to establish a unified safety level that corresponds to the original constraint settings.
>
> We can give an example from the locomotion tasks, which have three constraints regarding body balance, CoM height, and feet timing.
> Since the body balance and the CoM height significantly affect the robot's stability, it is necessary to increase the safety level of the corresponding constraints.
> However, the constraint on the feet timing is to prevent the robot from standing still and has little relation to safety.
> Thus, if the level of this constraint is increased, the robot control may become unstable.
> Even though the integrated cost function can be defined as the result of the "and" operation of all cost functions, finding a proper safety level is tricky.
> For this reason, it is difficult to conduct the proposed experiment.

---

> > ### Comment · Reviewer_SPhH · 2022-11-17
> > **Thanks...**
> >
> > ...for the clarifications and updates. I will carefully read the new version of the paper and consider increasing my score.

---

> > > ### Author Response · Authors · 2022-11-30
> > > **Further discussion**
> > >
> > > Dear Reviewer SPhH,
> > >
> > > Thanks again for your thoughtful review.
> > > We would like to ask if our answer and the modified paper improved clarification sufficiently.
> > > Any further questions or discussions are welcome.
> > >
> > > Sincerely,
> > >
> > > Paper authors.

---

> > > ### Author Response · Authors · 2022-12-12
> > > **Last day reminder**
> > >
> > > Dear Reviewer SPhH,
> > >
> > > Thanks again for reviewing our paper.
> > > Today is the last day of the review process, and we look forward to your response to the revised paper.
> > >
> > > Thank you,
> > >
> > > Paper authors.

---

> ### Author Response · Authors · 2022-11-13
> **Response to Reviewer SPhH (Part 1)**
>
> We thank reviewer SPhH for providing the opportunity to improve our manuscript with meaningful comments.
> We respond to reviewer SPhH's suggestions and comments below.
>
> ### **Q1. The paper is at parts very hard to follow.**
>
> Thanks for the comment, and we agree that some of the explanations in our paper could have been more intuitive.
> We have revised the structure and contents of the introduction section to convey our ideas clearly.
> We have also added rough overviews at the beginning of each section of the proposed method.
> Please see the revised version.
>
> ### **Q2. Comparison to "safe policy improvement".**
> > "computing the target distribution in an off-policy setting sounds a bit like ‘safe policy improvement’ (SPI), coined by Thomas et al."
>
> To the best of our knowledge, SPI seems to be quite different from the calculation of the proposed target distribution.
> We think that SPI and the proposed target distribution have nothing in common except that they use importance sampling.
> Both methods also have different purposes; SPI is to guarantee the performance of the policy over a certain level when updating the policy, and the target distribution is to train critics to estimate constraints or returns with low biases.
> If the question of the reviewer is intended to compare SPI and the safe policy update rule (we newly introduced this notion in the third paragraph of the introduction), they are also different in terms of the meaning of safety.
> In SPI, safety means guaranteeing that the performance of the updated policy is over a threshold. However, in the safe policy update rule, safety means that the updated policy should satisfy the safety constraints.
> If we misunderstood something, please let us know.
>
> ### **Q3. Is the overall methods ‘safe’ or reliable?**
>
> We will divide the initial policy into three cases and give an answer for each case.
> To clarify the definition of safety, We can say that a policy is safe or feasible if the policy satisfies all constraints.
> The first case is that the initial policy is safe. In that case, the policy keeps safe throughout training because the policy is updated in the feasible policy space using the trust region method.
> However, in practical cases, the policy may be unsafe sometimes due to the estimation error of the critic.
> The second case is that the initial policy is unsafe, but the feasible policy space is not empty.
> In that case, we can get a safe policy using the proposed gradient integration method within a finite time.
> Therefore, once we find a safe policy, it remains safe for the rest of the training.
> The last case is that the initial policy is unsafe and the feasible policy space is empty.
> As there is no solution for the safe RL problem in this case, there is no safe policy.

---

### Official Review · Reviewer_QThN · 2022-10-28

**Confidence:** 3
**Correctness:** 4
**Technical Novelty And Significance:** 3
**Empirical Novelty And Significance:** 2
**Recommendation:** 8

**Clarity, Quality, Novelty And Reproducibility:**

Clarity: clear but quite dense, some design choices are not well explained in the current state

Quality: good

Novelty: several elements are novel or their combination is novel (efficient TD($\lambda$), distributional critics, SAC-style surrogates, gradient integration).

Reproducibility: no code is available as far as I know, a computation cost analysis is missing

**Strength And Weaknesses:**

### Strengths

The paper involves technical elements that the authors explain in a clear way.
For instance, the introduction is super clear and provides a nice overview of the existing methods. In the background, the notations are clear and precise. The figures do a great job at facilitating the comprehension. The related work is exhaustive and well-organized.

The results on Safety Gym are quite nice, showing increasing or equal scores while reducing the number of violated constraints, which illustrates the empirical pertinence of the proposed method.

### Weaknesses

The paper is quite dense and I think this aspect could be improved. Some proofs might appear in the appendix instead.

On the title: use the term low-bias instead of low-biased?

On the RL method:
* It should be made clearer that the recursive formula (eq.7) is particularly useful to implement the method
* Fig.1 is nice, but a clarification as to why a projection is needed, i.e. different supports for distinct time-steps, would be welcome in the text
* why the off-policy version of TRPO? This choice should be motivated.
* Why TRPO over PPO? PPO enforces a trust region by modifying the policy gradient, while TRPO requires backtracking. If I understand correctly this is because backtracking includes the linearized constraints as well? This choice could be better motivated in the text.
* What is the cost of gradient integration in practice?

On experiments:
* There are many hyperparameters for the resulting method, how hard is it to hyperoptimize? I think a discussion on that aspect would benefit the work.


**Summary Of The Paper:**

The authors propose SDAC, a safe RL method based on trust regions that uses TD($\lambda$) for the distributional critic and SAC-style updates.
They provide an elegant and memory-efficient way to approximate TD($\lambda$).
They propose to perform gradient integration in order to avoid getting stuck due to unmet constraints.
The proposed method was tested on Safety Gym with very nice results.


**Summary Of The Review:**

This is a nice paper, featuring good writing and a genuine effort at making technical points digest to the reader.

The experiments are varied and convey a lot of information. They do a good job at validating the claims of the paper.

The proposed algorithm seems super heavy (k-distributional critics + backtracking + quadratic programming for gradient integration) but the computational cost is not quantified in this work, which I think is regrettable.

In the current state, it is somehow hard to see how the community will use this work if no open-source implementation and/or not more comments on computational cost and sensitivity of the hyperparameters.

For all these reasons, though the proposed method is inventive and sound, I only borderline recommend for acceptance. There are low hanging fruits that the authors could choose to address to increase the potential impact of the work.

---

> ### Author Response · Authors · 2022-11-13
> **Response to Reviewer QThN (Part 3)**
>
> ### **References**
>
> [1] Joshua Achiam. Spinning Up in Deep Reinforcement Learning. 2018.
>
> [2] Antonin Raffin, Ashley Hill, Adam Gleave, Anssi Kanervisto, Maximilian Ernestus, and Noah
> Dormann. Stable-baselines3: Reliable reinforcement learning implementations. Journal of
> Machine Learning Research, 22(268):1–8, 2021.
>
> [3] Siddhant Gangapurwala, Alexander Mitchell, and Ioannis Havoutis. Guided constrained policy
> optimization for dynamic quadrupedal robot locomotion. IEEE Robotics and Automation Letters,
> 5(2):3642–3649, 2020.
>
> [4] Schulman, John, et al. "High-dimensional continuous control using generalized advantage estimation." arXiv preprint arXiv:1506.02438 (2015).
>
> [5] Dabney, Will, et al. "Implicit quantile networks for distributional reinforcement learning." International conference on machine learning. PMLR, 2018.
>
> [6] Kim, Dohyeong, and Songhwai Oh. "Efficient Off-Policy Safe Reinforcement Learning Using Trust Region Conditional Value At Risk." IEEE Robotics and Automation Letters 7.3 (2022): 7644-7651.

---

> > ### Comment · Reviewer_QThN · 2022-11-17
> > **Response to authors' response**
> >
> > I want to thank the authors for the work done which IMO improved the current state of the paper.
> >
> > I still think that the choice of TRPO (Q5) and its off-policy version (Q4) could be made clearer in the text, and that they might trouble the reader, even in the revised version.
> >
> > I would also advocate for a cost analysis of SDAC against methods that integrate constraints.
> >
> > All in all, I am convinced by the work but I am still concerned about the presentation and justification of some key aspects of the method *in the paper*.
> >
> > I am still erring on the side of borderline acceptance for now.

---

> > > ### Author Response · Authors · 2022-11-18
> > > **Response to Reviewer QThN**
> > >
> > > Thanks for the response.
> > > To reflect the reviewer's concerns about the choice of TRPO over PPO (Q5) and its off-policy version (Q4), we have rewritten the beginning of Section 3.2.
> > > We have explicitly stated why we chose TRPO instead of PPO and why we used off-policy TRPO among several variants of TRPO.
> > > Please see the revised manuscript.

---

> > > > ### Comment · Reviewer_QThN · 2022-11-29
> > > > **Response to authors**
> > > >
> > > > Thanks for the revisions. I encourage authors to provide a cost analysis of SDAC against methods that integrate constraints. I am updating my score accordingly.

---

> ### Author Response · Authors · 2022-11-13
> **Response to Reviewer QThN (Part 2)**
>
> ### **Q6. What is the cost of gradient integration in practice?**
>
> We have added a discussion of the computational cost in Appendix E.
> The proposed gradient integration method has several subparts, and we will review each part's computational cost.
>
> First, we need to calculate policy gradients of each cost surrogate, $g_k$, and $H^{-1}g_k$ for $\forall k \in \{1, 2,..., K\}$.
> $H^{-1}g_k$ can be computed using the conjugate gradient method, which requires only a constant number of backpropagation on the cost surrogate, so the computational cost can be expressed as $K\cdot O(\mathrm{Backprop})$.
>
> Second, the quadratic problem in Section 3.3 is transformed to a dual problem, where the transformation process requires inner products between $g_k$ and $H^{-1}g_m$ for $\forall k,m \in \{1, 2,..., K\}$.
> The computational cost can be expressed as $K^2\cdot O(\mathrm{InnerProd})$.
>
> Finally, we solve the quadratic problem in the dual space, which is in $\mathbb{R}^{K}$, using some quadratic programming solvers.
> Since $K$ is usually much smaller than the number of policy parameters, the computational cost almost negligible compared to the others.
>
> As a result, we can conclude the cost of the gradient integration is $K\cdot O(\mathrm{Backprop}) + K^2\cdot O(\mathrm{InnerProd}) + C$.
> In our case, the number of constraints $K$ is one in the Safety Gym tasks and three in the locomotion tasks, so the practical computation time was not so long.
> We have logged the total training time of the Cassie task, including simulation time and update time, for the proposed method, WCSAC, and the traditional RL baselines, PPO and TQC (which do not use constraints).
> We have extended WCSAC to a multi-constrained version by adding additional Lagrange multipliers and the multipliers' optimizers to obtain training time.
> However, due to the sensitive hyperparameters for the multipliers (multiplier's damping coefficients, learning rates, and the initial multiplier values), the performance of WCSAC has not yet been optimized, so we did not add the WCSAC results to the manuscript.
> The results are presented below, where the training time is obtained by averaging the execution times of five random seeds.
> |Cassie Task| **SDAC** (proposed, safe RL) | WCSAC (safe RL) | PPO (RL) | TQC (RL) |
> |-|-|-|-|-|
> | Training time (hour) | 15.06|60.99|3.97|8.71|
>
> ### **Q7. Hyperparameters.**
> > "There are many hyperparameters for the resulting method, how hard is it to hyperoptimize?"
>
>
> - **[Discount factor, critic learning rate]** Since these are commonly used hyperparameters, we will omit them.
> - **[Trace-decay, trust region size]** We experimented with these hyperparameters and presented the results in Appendix D.4.
> From the results, we recommend setting the trace-decay to 0.95~0.99 as in other TD($\lambda$)-based papers [4].
> Also, the results show that the performance is not sensitive to the trust region size.
> However, if the trust region size is too large, the approximation error increases, so it is better to set it below 0.003.
> - **[Entropy coefficient]** We fixed this value in our experiments, but it can be adjusted automatically as done in SAC.
> - **[The number of atoms]** Although we did not experiment with the number of atoms, performance is expected to increase as the number of atoms increases, as in other distributional RL papers [5].
> - **[Length of replay buffer]** The effect of the length of the replay buffer on safe RL can be confirmed through the experimental results of [6], an off policy-based safe RL method.
> According to that paper, it does not impact performance unless it is too short. We recommend setting it to 10 to 100 times the collected trajectory length.
> - **[Constraint conservativeness $\alpha$, limit value $d_k$]** If the cost sum follows a Gaussian distribution, the mean-std constraint becomes the CVaR constraint. Then, the probability of the worst case can be controlled by adjusting $\alpha$.
> For example, if we set $\alpha=0.125$ and $d=0.03/(1-\gamma)$, the mean-std constraint makes the probability that the average cost is less than 0.03 during an episode greater than $95\%=\Phi(\phi(\Phi^{-1}(\alpha))/\alpha)$.
> Through this meaning, proper $\alpha$ and $d_k$ can be found.
> - **[Slack coefficient $\zeta$]** As mentioned at the end of Section 3.3, it is recommended to set this value as large as possible. Since $d_k - \zeta$ should be positive, we recommend setting $\zeta$ to $\min_k d_k$.
> - **[Reward weights]** These are used when defining the reward function for traditional RL methods, so these are not hyperparameters of our method.
>
> In conclusion, most hyperparameters are not sensitive, so there is no need to optimize. It seems that only $\alpha$ and $d_k$ need to be set based on the meaning described above. If the approximation error of critics is significant, the trust region size can be set smaller.
>
> ### **Q8. Reproducibility.**
>
> We have attached the source code in the supplementary material, so please refer to the code.

---

> ### Author Response · Authors · 2022-11-13
> **Response to Reviewer QThN (Part 1)**
>
> We thank reviewer QThN for the positive feedback and thorough review on our work.
> We respond to reviewer QThN's suggestions and comments below.
>
> ### **Q1. The paper is quite dense.**
>
> Thanks for the comments, and we agree with this.
> We have modified the structure and contents of the introduction section to make it easier to understand the proposed ideas and added summaries at the beginning of each section of the proposed method.
> Please see the revised version.
>
>
> ### **Q2. Title modification.**
> > "Use the term low-bias instead of low-biased?"
>
> Thanks for the recommendation; we have revised the title.
>
> ### **Q3. TD($\lambda$) target distribution.**
> > "It should be made clearer that the recursive formula (eq.7) is particularly useful to implement the method. Fig.1 is nice, but a clarification as to why a projection is needed."
>
> Thanks for pointing it out.
> To give technical details, we have provided the pseudocode of the recursive procedure for calculating the proposed target distribution in Appendix A.2.
> Also, we have modified the details of Figure 1 and added explanations in the caption to increase clarity.
>
> Here, we would like to give a simple explanation of the projection in Figure 1.
> If we do not perform the projection, the target distribution becomes $\hat{Z}_t^{\mathrm{tot}}$, which has $M + M'$ atoms, as shown in Figure 1.
> Then, the target distribution at $t-1$ step will have $2M+M'$ atoms, and recursively, the target at $t=1$ step will have $TM+M'$ atoms, which requires lots of memory.
> To prevent such memory inefficiency, we projected the target of all steps to have the same number of atoms.
>
> ### **Q4. Why the off-policy version of TRPO?**
>
> Thanks for the comment, we have added an explanation to the Section 3.2.
> We derive the novel SAC-like surrogate to increase the sample efficiency of the trust region-based methods. (The reason why the SAC style is chosen is that The performance of SAC outperforms the performance of TRPO, as observed in several RL benchmarks [1, 2].)
> Since SAC is an off-policy RL method, an off-policy version of TRPO is used instead of the original TRPO.
> Also, it can be seen that off-policy TRPO is used since off-policy algorithms generally have better sample efficiency than on-policy algorithms.
>
> ### **Q5. Why TRPO over PPO?**
> > "PPO enforces a trust region by modifying the policy gradient, while TRPO requires backtracking. If I understand correctly this is because backtracking includes the linearized constraints as well? This choice could be better motivated in the text."
>
> The reviewer has raised an interesting point about why we did not use PPO instead of TRPO.
> In general, PPO cannot calculate a policy gradient which reflect constraints.
> To reflect the constraints, PPO can be combined with the Lagrangian method, which converts the safe RL problem to a dual problem using Lagrange multipliers. Then, the policy gradient can be calculated by solving the dual problem using PPO, as in [3].
> However, in this case, PPO is considered a Lagrangian method. Since we argued in the introduction section that the trust region method has more advantages than the Lagrangian method, TRPO is applied to the proposed method.

---

### Author Response · Authors · 2022-11-13
**General Response to All Reviewers**

We would like to express our sincere gratitude to the reviewers for their valuable suggestions on how to improve our manuscript.
We have modified the manuscript according to the following common comments.

## **Q1. Motivation.**
- **[reviewer FqLh]** Some of the contributions are not well-motivated.
- **[reviewer QThN]** why the off-policy version of TRPO?

Like the reviewers' comments, we agree that some motivation for the contributions was ambiguous.
In summary, our contributions are threefold: we propose 1) a low-biased target distribution for distributional critics, 2) a gradient integration method to handle infeasible starting cases under multiple constraint settings, and 3) a SAC-style surrogate, which increases sample efficiency of the trust region method.

In the introduction section, we have explained why distributional critics are required.
This is because the mean-std constraint is better than the mean constraint for better safety in the stochastic RL setting.
The distributional critic is desirable for the mean-std constraint since estimating mean-std requires knowing the distributional shape of the sum of costs.
Then, we have modified the explanation of why critics should be trained with low biases and why trust region-based methods are required to solve the safe RL problem.
Also, we have emphasized that the issues of trust region-based methods, a lack of handling methods for multiple constraints and low sample efficiency, can be solved by the proposed gradient integration method and SAC-style surrogates.
Finally, an explanation of why off-policy TRPO is used to derive the SAC-style surrogates have been added in Section 3.2.

## **Q2. Clarification.**
- **[reviewer QThN]** Fig.1 is nice, but a clarification as to why a projection is needed.
- **[reviewer SPhH]** I had to read the introduction at least three times to have an idea of the approach, and the tense technical description does not help me.
- **[reviewer 97Q2]** I don't understand what figure 1 is trying to say, could the author be more clear?

As the reviewers pointed out, we have modified the structure and contents of the introduction section to deliver the main ideas intuitively.
Figure 1 shows the procedure for calculating the proposed target distribution, but for more clarification, we have added details to the figure and explanations to the caption.
Additionally, we have provided the pseudocode for calculating the proposed target distribution in Appendix A.2 to convey technical details.

## **Q3. Reproducibility.**
- **[reviewer QThN]** It is somehow hard to see how the community will use this work if no open-source implementation and/or not more comments on computational cost and sensitivity of the hyperparameters.
- **[reviewer 97Q2]** Based on the list of hyperparameters provided by the authors (and no other baseline details of the experimental parameters are provided), I think the reproducibility of the experimental results of this work is very low.
- **[reviewer FqLh]** The paper does not provide code but provides hyperparameters and settings in the appendix. Thus, reproducing the result might be possible but not easy.

To reflect the reviewers' comments on reproducibility, the source code has been attached to the supplementary material.

---

### Decision · Program_Chairs · 2023-01-20

**Decision:**

Reject

**Justification For Why Not Higher Score:**

The proposed method integrates several state-of-the-art techniques, but each technique has limited novelty.

**Justification For Why Not Lower Score:**

N/A

**Metareview: Summary, Strengths And Weaknesses:**

This paper proposes a trust-region based method of safe reinforcement learning (RL), which integrates several state-of-the-art techniques and is empirically shown to outperform existing safe RL methods.  More specifically, the proposed approach
1. learns distributional TD(\lambda) to balance the bias and variance of the estimated return, where a projection technique is used to keep the number of quantile points small,
2. uses Q-functions rather than advantage functions for SAC-like surrogates, and
3. uses gradient integration to deal with multiple constraint.
The experiments are conducted on safety gym environments with single constraint to compare against existing trust-region based safe RL methods that can deal with single constraints as well as locomotion tasks with multiple constraints to highlight the advantages of proposed method over standard (non-safe) RL that replaces the reward with the weighted sum of reward and safety.  Moreover, ablation study shows the effectiveness of each technique in the proposed approach.

The main strength of the paper is in the integration of state-of-the-art techniques in a way that it outperforms existing safe RL methods.  In particular, the advantages of the proposed approach over existing trust-region based safe RL methods are clearly demonstrated in the experiments.  Given the importance of safety in RL, the method that achieves the state-of-the-art performance is a valuable contribution.

The major weakness in the original submission was clarity (particularly introduction and experimental results), which however has been resolved satisfactory in the latest version of the paper.

One of the remaining weaknesses is in its limited novelty.  The distributional TD(\lambda) has also been studied in the prior work, as is also discussed in the paper.  While the projection technique may be new in the context of learning quantile-based distributional TD(\lambda), it is not particularly novel.  Also, the advantage of TD(\lambda) over Monte Carlo return is not so significant from the ablation study.  Q-functions are certainly used in many context of RL, while it is good to know that it is a particularly good choice in safe RL.  Gradient integration has some novelty, and its effectiveness over a naive method (with respect to the number of steps needed to satisfy all constraints) is clearly demonstrated, but the contribution by itself is limited.

Overall, weaknesses slightly outweigh strengths for this paper.  A suggestion for improvement is to empirically compare per-step computational cost of the proposed approach and baselines.  While the computational cost analysis was added in Appendix E, quantitative differences are unclear.